# The Fundamental Limits of Neural Networks for Interval Certified Robustness

**Matthew Mirman**                                          *matthew.mirman@inf.ethz.ch*

**Maximilian Baader**                                    *maximilian.baader@inf.ethz.ch*

**Martin Vechev**                                            *martin.vechev@inf.ethz.ch*

*Department of Computer Science, ETH Zurich*

**Reviewed on OpenReview:** *https://openreview.net/forum?id=fsacLLU35V*

## Abstract

Interval analysis (or interval bound propagation, IBP) is a popular technique for verifying and training provably robust deep neural networks, a fundamental challenge in the area of reliable machine learning. However, despite substantial efforts, progress on addressing this key challenge has stagnated, calling into question whether interval analysis is a viable path forward.

In this paper we present a fundamental result on the limitation of neural networks for interval analyzable robust classification. Our main theorem shows that non-invertible functions can not be built such that interval analysis is precise everywhere. Given this, we derive a paradox: while every dataset can be robustly classified, there are simple datasets that can not be provably robustly classified with interval analysis.

## 1 Introduction

As neural networks are increasingly used in safety critical environments, ensuring their behavior with *formal verification* has become a highly active research direction (Liu et al., 2019a; Huang et al., 2020). Because neural networks are often too large for complete verification methods, incomplete analysis techniques are frequently employed (Gehr et al., 2018) – these can scale to larger models though may fail to prove a property that actually holds (as demonstrated in Fig. 1). Indeed, recent progress in constructing provable neural networks has been achieved thanks to leveraging incomplete methods, and particularly interval (box) bound propagation (Mirman et al., 2018). However, while many improvements to *provable defenses* have been published (Gowal et al., 2018; Zhang et al., 2018; 2020; Xiao et al., 2019; Wong et al., 2018; Liu et al., 2021; Boopathy et al., 2021; Shi et al., 2021; Xu et al., 2021) (most building on IBP), progress remains far from satisfactory: the state-of-the-art certified robust accuracy is roughly 60% on CIFAR10 (Balunovic & Vechev, 2020), compared to state-of-the-art standard accuracies of above 95%. The stagnation of progress in constructing provably robust neural networks has led to a fundamental theoretical question:



Do efficiently verifiable neural networks exist?

**(Fundamental Theoretical Question)**



The first result addressing this question was investigated by Baader et al. (2020) which proved an analog to the universal approximation theorem (Cybenko, 1989; Hornik et al., 1989) for interval-analyzable networks. Wang et al. (2020) further showed that two hidden layer networks could also be interval-analyzable approximators. Anonymous (2022) also demonstrated that training with interval propagation converges with high probability. While it is helpful to know that searching for networks which can be easily analyzed might not be futile, these results do not explain, and even contradict the provable training gap that is observed in practice. A preliminary negative result was shown by Wang et al. (2020): verifying the robustness of arbitrary neural networks in general and thus translating arbitrary neural networks into interval-analyzable forms is NP-hard.

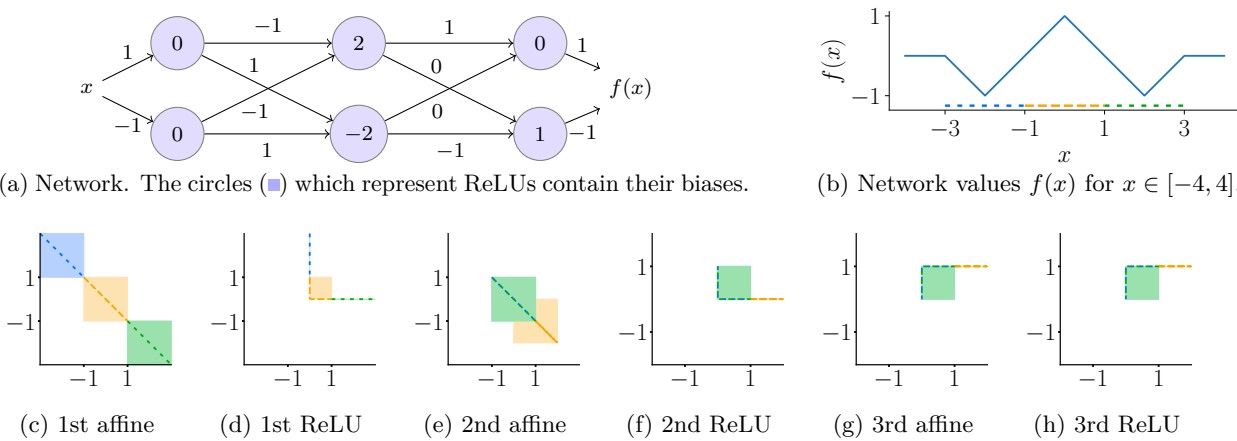

(a) Network. The circles (■) which represent ReLUs contain their biases.

(b) Network values $f(x)$ for $x \in [-4, 4]$.

(c) 1st affine    (d) 1st ReLU    (e) 2nd affine    (f) 2nd ReLU    (g) 3rd affine    (h) 3rd ReLU

Figure 1: An example of a neural network which is in fact robust, yet which interval arithmetic fails to *prove* is robust. The three intervals $L = [-3, -1]$ (■), $M = [-1, 1]$ (■) and $R = [1, 3]$ (■) are depicted using dashed lines in Fig. 1b. The interval propagation through the network is shown using the rectangles. The concrete values are shown by the dashed lines. After the 2nd affine layer (Fig. 1e) the orange and green box overlap while the orange lines do not overlap, demonstrating a loss of precision. The output Interval is $[-1, 1]$ for all three input intervals $L, M$ and $R$. In graphs (c) through (h) the vertical axis corresponds to the outputs of the upper set of neurons in (a) and the horizontal axis corresponds to the outputs of the lower set of neurons.

In our work, we provide a strong negative answer, thus explaining the provable training gap: we demonstrate that non-trivial datasets can not be classified by interval-provable networks.

Formally, given a neural network, or more generally any program, $f : \mathcal{X} \to \mathcal{Y}$, the goal of verification is to algorithmically prove that $f$ maps an *input spec* (specification), $S_I \subseteq \mathcal{X}$, to a subset of an *output spec*, $S_O \subseteq \mathcal{Y}$, where the *spec* $(S_I, S_O)$ is a member of a *spec-set* $\mathcal{S} \subseteq \mathcal{P}(\mathcal{X}) \times \mathcal{P}(\mathcal{Y})$. Interval analysis in particular replaces the basic operations of $f$ with interval arithmetic (Moore, 1966; Hickey et al., 2001), producing an interval extension, $f^{\#} :$ Intervals$(\mathcal{X}) \to$ Intervals$(\mathcal{Y})$, of $f$ such that every element of $S_I$ is mapped by $f$ to an element of $f^{\#}(S_I)$. As representing and computing intervals is efficient, $f$ is proven to meet the spec $(S_I, S_O)$ by proxy of proving $f^{\#}(\mathcal{S}_I) \subseteq S_O$. Here we consider *interval robustness classification specs* where the input specs are closed intervals, and the output specs are either $\mathbb{R}_{>0}$ or $\mathbb{R}_{<0}$.

**Main contributions.** In this paper, we present the first proofs capturing key limitations (incompleteness) of using ReLU based neural networks to build robust classifiers that can be certified with interval analysis:

- **Fundamental Imprecision of Interval Analysis (Theorem 4.10)**: In this theorem we identify for the first time the highly general conditions under which interval analysis looses precision. Specifically, we show that non-invertible functions can not be built using affine transformations and ReLUs such that over-approximation using interval analysis is precise everywhere. To make proving this theorem possible, we need to keep track of a strict interior relationship even when the dimension of the network increases in irrelevant ways. To do this, we develop the relative subset relation (Definition 4.2) and prove various useful properties in Section 4.1.

- **Impossibility of Interval-Provably Robust Classifiers (Section 5)**: Here we show that classifiers in particular can not be built such that they are provable robust with interval analysis in two key ways. We first demonstrate in Corollary 5.2 how this theorem can be immediately applied to show that it is impossible to construct a feed-forward ReLU-neural network of any shape (e.g., residual, convolutional, dense, fully-connected) that is *perfectly provably robust* (Definition 5.1) with interval analysis for even an unrealistically simple dataset with only three points. Informally, *perfect* here describes the property that the spec-set is maximal in that it includes as many *valid* robustness specs describing the dataset as possible without including any input specs which might render the spec-set unsatisfiable, including those that touch the shared border between two points. We then show in Theorem 5.8 that even when the requirement for perfect provability is relaxed to regions that are distant from each other ($\alpha$-interval provable with $\alpha < 1$ as in Definition 5.3), there are datasets with $\mathcal{O}(\alpha^{-1})$ points that can not be provably robustly classified with one-hidden layer networks using interval analysis.

- **Possibility of Interval-Agnostic Robust Networks (Proposition 5.10 and 5.11)**: Here we demonstrate the paradox that in both of the previous two cases, *perfectly robust classifiers* (Definition 5.1) can always be constructed, even if they are not necessarily provably robust using interval analysis. In the case of 1-d datasets, these can furthermore be constructed to be one-layer deep. Together with Theorem 5.8 and Corollary 5.2 this implies that the restriction that a network be analyzable with interval-arithmetic is severely limiting.

## 2 Problem Motivation

Studying the robustness of artificial neural networks has become an important area of research as they are increasingly deployed in safety-critical applications such as self-driving cars (Bojarski et al., 2016). Szegedy et al. (2013) first demonstrated that neural networks classifying images can be fooled into misclassification by imperceptible pixel changes in an otherwise correctly classified image.

Many of these fooling techniques, known as adversarial attacks, have been developed (Carlini & Wagner, 2017; Goodfellow et al., 2015; Kurakin et al., 2016; Shaham et al., 2015; Croce & Hein, 2019b; Papernot et al., 2016a; Wong et al., 2019b). To defend against these attack, methods strengthening models have been proposed (Papernot et al., 2016b; Tramèr et al., 2017; Wong et al., 2019a; Stutz et al., 2020; Bastani et al., 2016; Croce & Hein, 2020). A particular line of research aims to provide *formal* guarantees (i.e., verify) that neural networks behave correctly (Katz et al., 2017; Singh et al., 2018; 2019; Boopathy et al., 2019; Liu et al., 2019b; Wang et al., 2018; Balunovic et al., 2019; Zhang et al., 2021; Lin et al., 2021; Croce & Hein, 2019a; Croce et al., 2018). As complete verification of a neural network is NP-Hard (Katz et al., 2017), the majority of modern techniques are incomplete and based on over-approximating the behavior of a network (Gehr et al., 2018). While incomplete methods can be highly efficient, a correct classification of a network *might not be provably* correct, as illustrated in Fig. 1. In fact, for naturally trained neural networks, only a small percentage of non-attackable input images are verifiable.

To improve verification rates, techniques to train networks amenable to verification (Raghunathan et al., 2018; Mirman et al., 2018; Wong & Kolter, 2018; Wong et al., 2018) have been developed. While this has been an active area of research, the state-of-the-art developed by Balunovic & Vechev (2020) achieves a certified robust accuracy of 60.5% on CIFAR10, which is low given that state-of-the-art standard accuracy is over 95%.

The recent plateau of progress in closing this gap has raised concerns on whether there are theoretical limitations to neural network analysis (Salman et al., 2019). In this work, we prove fundamental limitations, which help explain the significant gap between certified robust accuracy, and standard accuracy. We focus on interval analysis, as some of the most successful and widely used methods are based on this relaxation (Mirman et al., 2018; Gowal et al., 2018).

## 3 Background

We now introduce key concepts and notation, a reference to which can be found in Appendix C.

### 3.1 General Notation

We begin with some general notation. For some positive natural $k \in \mathbb{N}$ we write $[k] := \{1, \ldots, k\}$. Given sets $T_1, \ldots, T_k$ and a set $Y \subseteq T_1 \times \ldots \times T_k$ we write $Y|_i$ for the restriction to the dimension $i$, or $Y|_i := \{y_i : y \in Y\}$.

Many of our proofs require spaces with slightly richer structure than $d$-dimensional real vector spaces. We will call these real binary-tree tensor-spaces, the class of which can be defined inductively as $\mathscr{T} = \{T_1 \times T_2 : T_1, T_2 \in \mathscr{T}\} \cup \{\mathbb{R}\}$. Given a space $S \in \mathscr{T}$, we can inductively define the set $\mathscr{B}^S$ of closed, non-empty, axis-aligned boxes over $S$: If $S = T_1 \times T_2$ for $T_1, T_2 \in \mathscr{T}$ then $\mathscr{B}^S = \{B_1 \times B_2 : B_1 \in \mathscr{B}^{T_1}, B_2 \in \mathscr{B}^{T_2}\}$ and if $S = \mathbb{R}$ then $\mathscr{B}^S = \{[a, b] | a, b \in \mathbb{R} \wedge a < b\}$ or $\mathscr{B}^S = \mathbb{I}$, the set of real intervals.

In cases where this richer tree structure is unnecessary, we consider $\mathbb{R}^d \in \mathscr{T}$ for $d \in \mathbb{N}_{>1}$ to be a real binary-tree tensor space by assuming a canonical ordering: $\mathbb{R}^d := \mathbb{R} \times \mathbb{R}^{d-1}$. We abuse notation and say $\mathscr{B}^d$ is specifically the set of closed, non-empty, axis-aligned boxes on the $d$-dimensional real vector space. We note that $\mathscr{B}^{\mathbb{R}^d} = \mathscr{B}^d$.

We write the box with center $c \in \mathbb{R}^d$ and radius $r \in \mathbb{R}^d_{\geq 0}$ as $\mathcal{B}_r(c) \coloneqq \{x \colon \forall i \in [d]. \exists \xi \in [-1,1]. x_i = c_i + \xi r_i\}$. We also write $\mathcal{B}^{\circ}{}_{\epsilon}(x) \coloneqq \{y \in \mathbb{R}^d \colon ||x - y||_{\infty} < \epsilon\}$ for $x \in \mathbb{R}^d$ and $\epsilon \in \mathbb{R}_{>0}$ for an open box. For a given box $B \in \mathscr{B}^d$ let $\mathcal{C}(B)$ denote its center and $\mathcal{R}(B)$ denote its radius vector such that $B = \mathcal{B}_{\mathcal{R}(B)}(\mathcal{C}(B))$.

For any bounded and non-empty set $C \subset S$ for $S \in \mathscr{T}$, the $l_{\infty}$-*hull*, written $\mathcal{H}_{\infty}(C)$, is the smallest axis aligned box containing $C$, which we formally define inductively: $\mathcal{H}_{\infty}(C) \coloneqq \mathcal{H}_{\infty}(C|_1) \times \mathcal{H}_{\infty}(C|_2)$ if $S = S_1 \times S_2$ for $S_1, S_2 \in \mathscr{T}$ and $\mathcal{H}_{\infty}(C) \coloneqq [\inf(C), \sup(C)]$ if $S = \mathbb{R}$.

For any set $S$ we write $\mathcal{P}(S)$ to mean the powerset of $S$. If $f \colon A \to A'$ and $S \subseteq A$ we write $f[S]$ or $f^{\mathcal{P}}(S)$ to mean $\{f(s) \colon s \in S\}$, but sometimes abuse notation and write $f(S) \coloneqq f[S]$ to avoid clutter. Similarly, we also occasionally write $f^{-1} \circ g^{-1}$ even when $f$ and $g$ are non invertible to mean $(g \circ f)^{-1}$.

## 3.2 Robustness and Interval Certification (IBP)

Suppose $f \colon \mathbb{R}^d \to \mathbb{R}$ is some function (i.e., neural network). We say that this network assigns a label $l \in \{-1, 1\}$ to a point $x \in \mathbb{R}^d$ if sign $f(x) = l$. In our case, we discuss $l_{\infty}$-adversarial region specifications. In this case, we say that $f$ is $\epsilon$-*robust around* $x$ with *label* $l$ if $\forall x' \in \mathcal{B}_{\epsilon}(x). f(x') = l$.

The goal of robustness certification is to provide a guarantee that a neural network is robust at some point. However, an analysis not producing a proof of robustness at a point does not mean that the neural network is *non*-robust at that point. This leads to efficient methods in terms of *over-approximation*, originally described as *abstract-interpretation* (Cousot & Cousot, 1977) and applied to neural networks by Gehr et al. (2018).

**Interval Analysis**   In this paper, we focus on the *Interval (or Box)* relaxation, and in particular, Interval Bound Propagation (IBP) (Gowal et al., 2018), also known as interval analysis, or the Interval domain for abstract-interpretation. An interval, $B \in \mathscr{B}^d$, can either be represented as a center $c \in \mathbb{R}^d$ and radius $r \in \mathbb{R}^d_{\geq 0}$ as before, or as a lower-bound and upper bound, $\iota_l, \iota_u \in \mathbb{R}^d$ respectively such that for each dimension $j \in [d]$, we have $\iota_{l,j} \leq \iota_{u,j}$. The two representations are related as follows: $\iota_l = c - r$ and $\iota_u = c + r$, or $c = \frac{1}{2}(\iota_l + \iota_u)$ and $r = \frac{1}{2}(\iota_u - \iota_l)$.

**Analyzing Neural Networks**   The application of IBP to neural networks with ReLU-activations is straightforward. In this paper, we consider (feed forward) neural networks defined inductively as follows:

**Definition 3.1.** A $\sigma$-(neural) network with $\sigma$-activations is a program, $f$, inductively defined by the grammar $f ::= f \circ f | \langle f, f \rangle | \mathsf{Sum} | \mathsf{Dup} | \kappa | \kappa' \cdot | \sigma$ for $\kappa, \kappa' \in \mathbb{R}$ and $\kappa' \neq 0$, and interpreted as a function with domain $I \in \mathscr{T}$ and codomain $O \in \mathscr{T}$ as follows:

- *Sequential Computation:* Given $f = g_1 \circ g_2$ and $g_1, g_2$ are interpreted as functions $g_1 \colon B \to O$ and $g_2 \colon I \to B$, we say $f \colon I \to O$ computes $f(x) = g_1(g_2(x))$.

- *Parallel:* Given $f = \langle g_1, g_2 \rangle$ and $g_1, g_2$ are interpreted as functions $g_1 \colon I_1 \to O_1$ and $g_2 \colon I_2 \to O_2$ and we say $f \colon I_1 \times I_2 \to O_1 \times O_2$ computes $f(x) = (g_1(x_1), g_2(x_2))$.

- *Addition:* Given $f = \mathsf{Sum}$ we say $f \colon \mathbb{R}^2 \to \mathbb{R}$ computes $f(x) = x_1 + x_2$.

- *Duplication:* Given $f = \mathsf{Dup}$, we say $f \colon I \to I^2$ computes $f(x) = (x, x)$.

- *Constant:* Given $f = \kappa$ we say $f \colon \mathbb{R} \to \mathbb{R}$ computes $f(x) = \kappa$.

- *Multiplication by a Constant:* Given $f = \kappa \cdot$ we say $f \colon \mathbb{R} \to \mathbb{R}$ computes $f(x) = \kappa \cdot x$.

- *Activation:* Given $f = \sigma$ we say $f \colon \mathbb{R} \to \mathbb{R}$ computes $f(x) = \sigma(x)$.

We note that duplication and addition are the only relational operations here.

For the purposes of exploring its limits, we view IBP as a method that implicitly constructs a transformed function which acts on intervals. We describe this transformed function inductively as well:

**Definition 3.2 (Interval Analysis).** The *interval transformation*, $f^{\#}$, of a ReLU-network $f$ is as follows:

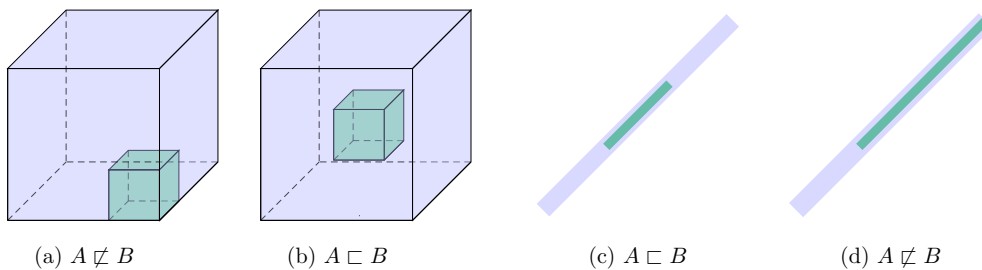

(a) $A \not\sqsubset B$        (b) $A \sqsubset B$        (c) $A \sqsubset B$        (d) $A \not\sqsubset B$

Figure 2: Visualization of the relative interior relation. Green (■) boxes are $A$ and purple (■) boxes are $B$.

- *Sequential Computation:* If $f = g_1 \circ g_2$ then $f^{\#}(B) \coloneqq g_1^{\#}(g_2^{\#}(B))$.

- *Parallel:* If $f = \langle g_1, g_2 \rangle$ then $f^{\#}(B) \coloneqq g_1^{\#}(B|_1) \times g_2^{\#}(B|_2)$.

- *Addition:* If $f = \mathsf{Sum}$ then $f^{\#}(B) \coloneqq \{a + b \colon a \in B|_1 \wedge b \in B|_2\}$.

- *Duplication:* If $f = \mathsf{Dup}$ then $f^{\#}(B) \coloneqq B \times B$.

- *Constant:* If $f = \kappa$ for $\kappa \in \mathbb{R}$ then $f^{\#}(B) \coloneqq \{\kappa\}$.

- *Multiplication by a Constant:* If $f = \kappa\cdot$ for some non-zero $\kappa \in \mathbb{R}$, then $f^{\#}(B) \coloneqq \{\kappa \cdot x \colon x \in B\}$.

- *Activation:* If $f = \mathrm{ReLU}$, then $f^{\#}(B) \coloneqq \{\mathrm{ReLU}(x) \colon x \in B\}$.

**Proposition 3.3.** The interval transformer, $f^{\#}$, over-approximates $f$, meaning that $f[B] \subseteq f^{\#}(B)$.

## 4 Fundamental Imprecision of Interval Analysis

Here we show our main result, that no neural network can be perfectly provably robust with interval analysis for simple functions. We first introduce the necessary machinery (in particular, the concept of relative interior outlined in Fig. 2) and lemmas that allow us to show a relationship between whether the network represents an invertible function, and where there is an approximation error.

Counterintuitively, rather than being able to show that the transformed network is imprecise for a specific input box (i.e., that for a specific box $B$, we know $f[B] \subsetneq f^{\#}(B)$), we must, for any input box $B$ containing non-invertible points on its surface, *find* an input box, $A$, that is a strict subset of $B$ (via a particular notion of strict defined below), such that $f[B] \subseteq f^{\#}(A)$. The fact that $A$ is a strict subset of the relative interior of $B$ implies that interval analysis is imprecise enough on the network such that it can not be used to prove desired properties of $B$ (such that $f$ is perfectly robust for $B$). It is however crucial that $A$ not be required to be *too* strict a subset of $B$. One might be tempted to find subsets of the topological interior of $B$. This however leads to significant technical issues: we need to have a notion of strict subset that applies even when some of the neurons in the network are unused (and zero). One can imagine the set representing the possible activations of those neurons as a lower dimensional surface embedded in a higher dimensional space, as in the case of Fig. 2(c) and (d). In this case, the interior of $B$ would be empty, even though we might have identified a subset of it that induces imprecision.

### 4.1 The Relative Subset Relation

We begin by formalizing the intuitive concept from Fig. 2 using the known notion of *relative interior*, and demonstrating some useful lemmas related to it. First, recall for a set $S \subseteq \mathbb{R}^d$ that the *affine hull* of $S$, written $\mathrm{aff}(S)$ is the smallest linear-subspace of $\mathbb{R}^d$ that contains $S$.

**Definition 4.1.** We recall the standard definition of *relative interior*:
$$\mathrm{relint}(S) \coloneqq \{x \in S \colon \exists \epsilon > 0.\, \mathcal{B}^{\circ}{}_{\epsilon}(x) \cap \mathrm{aff}(S) \subseteq S\}.$$

We note that if $S \in \mathscr{B}^d$, we can restate the relative interior as $\mathrm{relint}(S) = \{x \in S \colon \forall i \in [d].\, x_i \in S|_i^\circ \cup \{\mathcal{C}(S)_i\}\}$, where $S|_i^\circ$ is the interior of $S$'s restriction to dimension $i$.

**Definition 4.2** *(Relative Subset).* $A$ is a *relative subset* of $B$, written $A \sqsubset B$, if and only if $A \subseteq \mathrm{relint}(B)$.

We note that if $A, B \in \mathscr{B}^d$, we can rephrase $A \sqsubset B$ as follows: $A \subseteq B$ and for each dimension, $i$, where $B|_i^\circ$ is not empty, $A|_i \subseteq B|_i^\circ$ or more concisely, $\forall i \in [d].\, (B|_i^\circ \neq \emptyset \implies A|_i \subseteq B|_i^\circ)$. In particular, in one dimension, for real intervals $[a, b]$ and $[a', b']$ we have $[a, b] \sqsubset [a', b']$ if and only if $a' < a \leq b < b'$ or $a = a' = b' = b$.

Let $A, A', B, B', C$ be bounded and non-empty subsets of $\mathbb{R}^d$ in the following lemmas (the proofs of which can be found in Appendix B.1):

**Lemma 4.3** *(Respects Projection).* $A \sqsubset B$ implies $A|_i \sqsubset B|_i$.

**Lemma 4.4** *(Respects Cartesian Product).* $A \sqsubset B$ and $A' \sqsubset B'$ implies $A \times A' \sqsubset B \times B'$.

**Lemma 4.5** *(Downward Union).* $A \sqsubset C$ and $B \sqsubset C$ implies $A \cup B \sqsubset C$.

**Lemma 4.6** *(Downward Hull).* $C \in \mathscr{B}^d$ and $A \sqsubset C$ implies $\mathcal{H}_\infty(A) \sqsubset C$.

The following two lemmas are trivial and we frequently use them without mention:

**Lemma 4.7** *(Singleton Reflexivity).* $\{a\} \sqsubset \{a\}$.

**Lemma 4.8** *(Center-Singleton is Always a Relative Subset).* $A \in \mathscr{B}^d$ implies $\{\mathcal{C}(A)\} \sqsubset A$.

It is important to note that some simple related properties counterintuitively do not always hold. Namely, if $A \sqsubset B$ and $B \subseteq C$ it is not always the case that $A \sqsubset C$. Furthermore, if $A \sqsubset B$ and $A' \sqsubset B'$ it is not always the case that $\mathcal{H}_\infty(A \cup A') \sqsubset \mathcal{H}_\infty(B \cup B')$.

## 4.2 Inversion With Respect to The Relative Subset Relation

Here we demonstrate that neural networks can loosely invert sets with respect to the relative subset relation. More formally, for any neural network $f$ with ReLU-activations, one can essentially always find a strict subset, $X'$ of the relative interior of a box $X$ that the neural network maps to a superset of a specified subset $Y$ of the relative interior of the $l_\infty$-hull of $f(X)$.

**Lemma 4.9** *(Concrete Relative Inversion).* Suppose $f$ is a feed-forward network with ReLU-activations and $Y, X' \in \mathscr{B}^d$ and $X$ is compact and non-empty. Then
$$Y \sqsubset \mathcal{H}_\infty(f(X)) \implies \exists X' \sqsubset \mathcal{H}_\infty(X).\, Y \subseteq f^\#(X').$$

PROOF OVERVIEW. *(Full Proof in Appendix B.2)* The proof is by structural induction on the construction of $f$. We use the lemma itself as the induction hypothesis. Below we outline three key cases of the structural induction: sequential computations, relational parallel computations, and ReLU:

*Case:* $f = g \circ h$. *(Sequential Computation)*

*Subproof.* By definition, $Y \sqsubset \mathcal{H}_\infty(g \circ h(X))$. Thus, there is some $H \sqsubset \mathcal{H}_\infty(h(X))$ such that $Y \subseteq g^\#(H)$ by the induction hypothesis on $g$. Applying the induction hypothesis again with the network $h$, we get a set $X' \sqsubset \mathcal{H}_\infty(X)$ such that $H \subseteq h^\#(X')$. Thus, $Y \subseteq g^\#(H) \subseteq g^\# \circ h^\#(X') = f^\#(X')$. ◁

*Case:* $f = \mathsf{Dup}$. *(Duplication)*

*Subproof.* We know $Y|_1 \sqsubset \mathcal{H}_\infty(X)$ and $Y|_2 \sqsubset \mathcal{H}_\infty(X)$ by Lemma 4.3. By Lemma 4.5 and Lemma 4.6, we know $X' := \mathcal{H}_\infty(Y|_1 \cup Y|_2)$ is a relative subset of $\mathcal{H}_\infty(X)$, and thus $Y \subseteq X' \times X' = f^\#(X')$. ◁

*Case:* $f = \langle g_1, g_2 \rangle$. *(Parallel)*

Figure 3: A visualization of the claims of Theorem 4.10. The neural network, $f$ classifies three inputs $x = -2, 0, 2$ to respective labels $y = -1, 1, -1$ as in Corollary 5.2. Here, $y = 0$ is the non-invertability and $f^{-1}(y) = \{-1, 1\}$. These two points mapping to $y$, which constitute $N \subseteq f^{-1}(y)$, are marked in red (■). The $l_\infty$-hull, $\mathcal{H}_\infty(N)$, is marked in green (■). The perfect approximation of $f$ on $\mathcal{H}_\infty(N)$ (i.e., $\{(v, f(v)) \colon v \in N\}$) is marked in blue (■). We can see that the interval approximation of the relative interior box $M \sqsubset \mathcal{H}_\infty(N)$ looses precision by looking at the orange region (■).

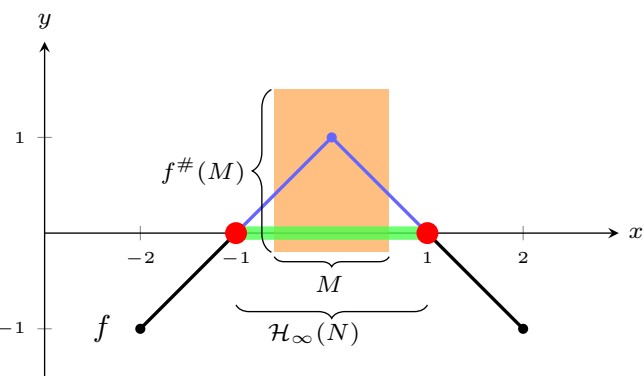

*Subproof.* We know both that $Y|_1 \sqsubset \mathcal{H}_\infty(g_1(X|_1))$ and $Y|_2 \sqsubset \mathcal{H}_\infty(g_2(X|_2))$, so we can apply the induction hypothesis twice to produce $L \sqsubset \mathcal{H}_\infty(X|_1)$ and $R \sqsubset \mathcal{H}_\infty(X|_2)$ such that $Y|_1 \subseteq g_1^{\#}(L)$ and $Y|_2 \subseteq g_2^{\#}(R)$. We choose $X' = L \times R$. By Lemma 4.4, we have $X' \sqsubset \mathcal{H}_\infty(X)$. Then $Y \subseteq g_1^{\#}(X'|_1) \times g_2^{\#}(X'|_2) = f^{\#}(X')$. ◁

*Case: $f = \mathrm{ReLU}$.* *(Activation)*

*Subproof.* We know that $f \colon \mathbb{R} \to \mathbb{R}$, which simplifies the proof. In this case, $l_\infty$-hull of $f(X)$ must either be a subset of $\mathbb{R}_{\geq 0}$ so either $Y$ is the singleton set containing zero, or a subset of $\mathbb{R}_{>0}$. In the first case, we pick $X'$ to be an easy-to-pick (the singleton set containing center as in Lemma 4.8) relative subset of the hull of $X$. Otherwise, we can pick $Y$ itself, since $\mathrm{ReLU}(Y) = Y \subseteq \mathcal{H}_\infty(X)$. ◁

*Case: $f = c \cdot$ for $c \neq 0$.* *(Mult. by a Constant)*

*Subproof.* Let $X' = \mathcal{B}_{|c^{-1}|\mathcal{R}(Y)}(c^{-1}\mathcal{C}(Y))$. Then clearly, $Y \subseteq f(X')$. Because we know that $d = 1$ we can write $x_l = \inf X$, $x_u = \sup X$, $y_l = \inf Y$ and $y_u = \sup Y$. We note that $x_l = \mathcal{C}(\mathcal{H}_\infty(X)) - \mathcal{R}(\mathcal{H}_\infty(X))$. Similar expressions can be derived for $x_u, y_l$ and $y_u$. Supposing $c > 0$ (the other case is analogous) and $x_l < x_u$ (the proof is similar when equal), we have $cx_l < y_l \leq y_u < cx_u$ by $Y \sqsubset \mathcal{H}_\infty(f(X))$.

Then we know $x_l < |c^{-1}|y_l \leq |c^{-1}|y_u < x_u$, and thus $X' \sqsubset \mathcal{H}_\infty(X)$. ◁

*(Further Cases in Appendix B.2)*

## 4.3 Impossibility for Non-Invertibility

We now prove our central result, that non-invertible neural networks necessarily induce approximation imprecision. Essentially, as visualized in Fig. 3, we show that there is a box, $M$ which is a relative subset (i.e., usually very strict subset) of the $l_\infty$-hull of any region for which the network is entirely not injective, such that analyzing the network with $M$ includes any of the non-invertible points in the inferred approximation.

The key idea is that while there may be non-invertibilities on the boundary, $M$ does not include these, as it is a relative subset. Thus, we can use this theorem to infer areas where analyzing the network produces approximations that include points that are not in the concrete, or true, set of possible network outputs.

**Theorem 4.10** (*Fundamental Imprecision of Interval*). Suppose $f \colon S \to T$ with $S, T \in \mathcal{T}$ is a ReLU-network and $y \in \mathbb{R}^m$ and $N \subseteq f^{-1}(y)$ is compact and non-empty. Then assuming $M \in \mathscr{B}^S$:
$$\exists M \sqsubset \mathcal{H}_\infty(N). \, y \in f^{\#}(M).$$

PROOF OVERVIEW. *(Full Proof in Appendix B.2)* Again, the proof is by structural induction on the construction of $f$, using the theorem itself as the induction hypothesis. Below we outline three key cases: sequential computations, relational parallel computations, and addition:

*Case:* $f = g \circ h$                                                         *(Sequential Computation)*

*Subproof.* We know that $N \subseteq h^{-1} \circ g^{-1}(y)$, and thus can use the induction hypothesis on $h$ to produce $M' \sqsubset \mathcal{H}_\infty(h(N))$ such that $y \in g^\#(M')$. By Lemma 4.9, we know that there is some $M \sqsubset \mathcal{H}_\infty(N)$ such that $M' \subseteq h^\#(M)$ and thus that $y \in g^\# \circ h^\#(M)$.    ◁

*Case:* $f = \mathsf{Dup}.$                                                           *(Duplication)*

*Subproof.* $N = \{y_1\}$ because $N \subseteq \{y_1\} \cap \{y_2\}$. By singleton reflexivity, $N \sqsubset \mathcal{H}_\infty(N)$. Thus, $y \in f^\#(N)$.    ◁

*Case:* $f = \langle g_1, g_2 \rangle.$                                                          *(Parallel)*

*Subproof.* First we know $N|_1 \subseteq g_1^{-1}(y_1)$ and $N|_2 \subseteq g_2^{-1}(y_2)$ by projection and that $N|_1$ and $N|_2$ are still compact and non-empty. Thus, by the induction hypothesis twice we see that there are boxes $M_1 \sqsubset \mathcal{H}_\infty(N|_1)$ and $M_2 \sqsubset \mathcal{H}_\infty(N|_2)$ such that $y_1 \in g_1^\#(M_1)$ and $y_2 \in g_2^\#(M_2)$. Then $M_1 \times M_2 \sqsubset \mathcal{H}_\infty(N)$ by Lemma 4.4. Then $y_1 \in f^\#(M_1 \times M_2)|_1$ and $y_2 \in f^\#(M_1 \times M_2)|_2$ by soundness. Thus, there is some box $M \sqsubset \mathcal{H}_\infty(N)$ such that $y \in f^\#(M)$.    ◁

*Case:* $f = \mathsf{Sum}.$                                                            *(Addition)*

*Subproof.* Because $f \colon \mathbb{R}^2 \to \mathbb{R}$, we know $f^{-1}(y) = \{(a, y - a) \colon a \in \mathbb{R}\}$. We can pick $M = \{\mathcal{C}(\mathcal{H}_\infty(N))\}$ and demonstrate that $M \sqsubset \mathcal{H}_\infty(N)$ and that $y \in f^\#(M)$.    ◁

*Case:* $f = \mathrm{ReLU}.$                                                         *(Activation)*

*Subproof.* In this case, $y = \mathrm{ReLU}(x)$ can either be zero or greater than zero. If $y > 0$, then $N = \{y\}$ by the definition of ReLU, and thus that $\{y\} \sqsubset \mathcal{H}_\infty(N)$ and $y \in f^\#(\{y\})$. Otherwise, $y = 0$ and thus $N = (-\infty, 0]$. We thus know $\{\mathcal{C}(\mathcal{H}_\infty(N))\} \sqsubset \mathcal{H}_\infty(N)$ and finally $y \in f^\#(\{\mathcal{C}(\mathcal{H}_\infty(N))\})$.    ◁

*(Further Cases in Appendix B.2)*

## 5 The Paradox of Interval Provable Network Construction

In this section we first demonstrate a variety of unexpectedly and problematically simple scenarios (although Theorem 4.10 implies many more) that if appear in a dataset, imply the impossibility of constructing an interval-provably robust network. We then show the paradox that in each of these cases, and further for every dataset, it is possible to build a perfectly robust classifier without consideration of interval analysis.

### 5.1 Impossibility of Perfect Provable Robustness of Unconstrained Networks

Here we demonstrate an immediate application of Theorem 4.10: there are simple datasets that no neural network can perfectly (as defined below), and provably robustly with interval classify.

**Definition 5.1.** We say that a neural network $f \colon \mathbb{R}^d \to \mathbb{R}$ is a *perfectly $\nu$-(provable) robust classifier* with $\nu \in (0, 1]$ for points $x_1, \ldots, x_n \in \mathbb{R}^d$ and labels $l_1, \ldots, l_n \in \{-1, 1\}$ if given $\delta = \frac{1}{2} \min_{i \neq j} \|x_i - x_j\|_\infty$ then $\forall \epsilon < \nu \delta. \forall i \in [n]. f(\mathcal{B}_\epsilon(x_i)) l_i > 0$. If it is *provable* then we also have that $\forall i \in [n]. f^\#(\mathcal{B}_\epsilon(x_i)) l_i > 0$.

We note that the specification-set induced by this definition is perfect as described in the introduction, in that it includes every valid robustness specification describing the dataset $(x_1, l_1), \ldots, (x_n, l_n)$: every robustness specification $(S_{I,i}, S_{O,i})$ where $x_i \in S_{I,i}$ and $l_i \in S_{O,i}$ and $S_{I,i} \subsetneq \mathcal{B}_\delta(x_i)$ is part of the specification-set, but not robustness specifications for different (and possibly differently classified) points that would be touching.

**Corollary 5.2** (*Perfectly Provably Robust Classifiers are Impossible*)**.** There is no feed forward ReLU-network that is a perfectly 1-provably robust classifier for the dataset $D = \{(-2, -1), (0, 1), (2, -1)\}$.

PROOF. Suppose $f \colon \mathbb{R} \to \mathbb{R}$ is a perfectly 1-provably classifier for this dataset. Then by continuity we know that $f(-1) = f(1) = 0$, and thus that $\{-1, 1\} \subseteq f^{-1}(0)$ (which is a compact non-empty set). Then by application of Theorem 4.10, there is some set $M \sqsubset \mathcal{H}_\infty(\{-1, 1\})$ such that $0 \in f^\#(M)$. Rephrased, this

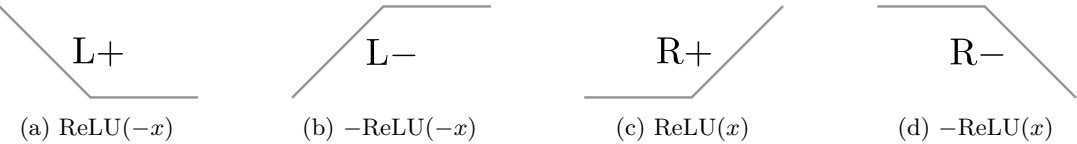

Figure 4: The orientations of neurons captured by the sets $L+$, $L-$, $R+$, and $R-$.

means that there are $a, b \in (-1, 1)$ such that $0 \in f^{\#}([a, b])$. This contradicts the definition of a provably 1-provable robust classifier however. $\square$

### 5.2 Impossibility of Imperfect Provable Robustness for One-Layer Networks

In this section we show a impossibility result for the case where the robust regions around inputs in the dataset do not overlap one another, and are thus imperfectly robust. To do this, we focus on single layer neural networks, as prior work shows possibility results here for deeper networks (Baader et al., 2020). Specifically, we present an upper-bound on the number of points that can be proven to be robustly classified with interval for a single-layer network. We do this by constructing a worst-case dataset, which we call *flips*. We begin by formalizing this dataset, and the notion of robust and provable robustness for a classifier on this dataset.

**Definition 5.3.** We say:

- A *d-flip* is a point $\hat{\mathbf{x}}_i \in \mathbb{R}^d$ with $\hat{\mathbf{x}}_{i,1} := 2i$ and $\hat{\mathbf{x}}_{i,j} = 0$ with *label* $\hat{1}_i := (-1)^i$. If $d$ is unspecified, it is taken to mean a 1-flip.

- $f : \mathbb{R} \to \mathbb{R}$ is a *classifier for $k$ flips* if $\forall i \in [k]. f(\hat{\mathbf{x}}_i) = \hat{1}_i$.

- $F : \mathscr{B} \to \mathcal{P}(\mathbb{R})$ is an *$\alpha$-classifier for $k$ flips* if $\forall i \in [k]. \forall y \in F([\hat{\mathbf{x}}_i - \alpha, \hat{\mathbf{x}}_i + \alpha]). \operatorname{sign} y = \hat{1}_i \wedge F(\{\hat{\mathbf{x}}_i\}) = \{\hat{1}_i\}$.

- If $f^{\mathcal{P}}$ (the perfect transformation) is an $\alpha$-classifier for $k$ flips, we say $f$ is an *$\alpha$-robust classifier* for $k$ *flips*.

- If $f^{\#}$ (the interval transformation from Definition 3.2) is an $\alpha$-classifier for $k$ flips, then we say that $f$ is a *provably $\alpha$-robust classifier* for $k$ *flips*.

We now specify the notion of single-layer network for which we demonstrate bounds:

**Definition 5.4.** A *single-layer $\sigma$-network*, $f : \mathbb{R} \to \mathbb{R}$, with *$n$-neurons* and *$\sigma$-activations* is a function with *pre-activation weights*, $N \in \mathbb{R}^n$, *pre-activation bias*, $b \in \mathbb{R}^n$, *post-activation weights*, $M \in \mathbb{R}^n$, and *post-activation bias*, $d \in \mathbb{R}$ (the weights and biases are known as *parameters*), such that $f(x) = M \cdot \sigma(Nx + b) + d$.

We note that while Definition 3.2 defines an ordering of addition, the above definition does not. While concrete-addition is associative, this does not necessarily hold for abstract addition. However, thankfully, for the interval transformation, it is, and the bounds we demonstrate apply to any ordering of the operations.

**Definition 5.5.** Given a single-layer ReLU-network, $f$ with $n$ neurons and pre/post-activation weights $N, M$, the absolute and standard *imprecision-contributions* (respectively) of $f$ at $x$ are:

$$\mathcal{A}_{D,S}(x) := \sum_{N_i x + b_i \geq 0 \wedge M_i \in S \wedge N_i \in D} |M_i N_i|,$$

$$\text{and } \mathcal{I}_{D,S}(x) := \sum_{N_i x + b_i \geq 0 \wedge M_i \in S \wedge N_i \in D} M_i N_i$$

where $D$ can be the set $L := \mathbb{R}_{\leq 0}$, the set $R := \mathbb{R}_{\geq 0}$ or $\mathbb{R}$ and $S$ can be $+ := \mathbb{R}_{\geq 0}$, the set $- := \mathbb{R}_{\leq 0}$ or $\mathbb{R}$.

Intuitively, $L+, L-, R+, R-$ correspond to the orientations that a neuron can take, as visualized in Fig. 4. $L$ (resp. $R$) results in contributions from neurons that activate as the argument $x$ of the imprecision-contribution function decreases (resp. increases). Note that $f'(x) = \mathcal{I}_{\mathbb{R},\mathbb{R}}(x)$ if the derivative of $f$ is defined at $x$.

**Lemma 5.6** *(End-Neuron Imprecision-Bound).* For all $\kappa \geq 0$ and single-layer ReLU-networks $f$ that classify $k$-flips for $k = \lceil \kappa \rceil + 5$, we have $\kappa < \max\{\mathcal{A}_{L,\mathbb{R}}(\hat{\mathbf{x}}_1), \mathcal{A}_{R,\mathbb{R}}(\hat{\mathbf{x}}_k)\}$.

PROOF OVERVIEW. We prove this by induction on $c := \lfloor \kappa \rfloor$, using two simultaneous inductive invariants:

$$c \leq \mathcal{A}_{L,+}(\hat{\mathbf{x}}_2) - \mathcal{A}_{R,+}(\hat{\mathbf{x}}_2) + \mathcal{A}_{R,+}(\hat{\mathbf{x}}_k) - \mathcal{A}_{L,+}(\hat{\mathbf{x}}_k),$$

and

$$c \leq \mathcal{A}_{L,-}(\hat{\mathbf{x}}_1) - \mathcal{A}_{R,-}(\hat{\mathbf{x}}_1) + \mathcal{A}_{R,-}(\hat{\mathbf{x}}_{k-1}) - \mathcal{A}_{L,-}(\hat{\mathbf{x}}_{k-1}).$$

This proof involves two key observations: (i) once imprecision-contribution in a direction has accumulated, it will only be larger for points further in that direction, (ii) one must measure not just the accumulated growth of the imprecision-contribution at the ends of the approximated data ($\hat{\mathbf{x}}_1$ and $\hat{\mathbf{x}}_k$) in the out-wards directed neurons, but the growth of the *relative* imprecision-contribution excluding contribution from in-wards directed neurons. We make these observations more precise in the full proof in the appendix.

*(Full proof in Appendix A)*

Before demonstrating the result for single layer networks, we require one further lemma, used to find a specific data-point with enough accumulated imprecision contribution to cause a violation:

**Lemma 5.7** *(Lower-Bound on Imprecision-Contribution).* For any $a \leq 1$ and $k \geq \lceil \frac{2}{a} \rceil + 5$ and single-layer ReLU-network $f$ that classifies $k$ flips, there is some point $j \in [k]$ such that

$$\hat{1}_j a^{-1} (f(\hat{\mathbf{x}}_j + a) + f(\hat{\mathbf{x}}_j - a))$$
$$< \mathcal{A}_{\mathbb{R},\mathbb{R}}(\hat{\mathbf{x}}_j) + \mathcal{A}_{R,\mathbb{R}}(\hat{\mathbf{x}}_j - a) + \mathcal{A}_{L,\mathbb{R}}(\hat{\mathbf{x}}_j + a).$$

PROOF OVERVIEW. By using $c := \frac{2}{a}$ we can apply Lemma 5.6 (with $\kappa = c$), to show bounds for either the left or right-most $\hat{\mathbf{x}}$ (i.e., $j = 1$ or $j = k$). For this point, we use the knowledge that the function is continuous, piecewise differentiable, to find points $l \in [\hat{\mathbf{x}}_j - a, \hat{\mathbf{x}}_j]$ and $u \in [\hat{\mathbf{x}}_j, \hat{\mathbf{x}}_j + a]$ such that $\frac{f(\hat{\mathbf{x}}_j) - f(\hat{\mathbf{x}}_j - a)}{a} < f'(l)$ and $f'(u) < \frac{f(\hat{\mathbf{x}}_j + a) - f(\hat{\mathbf{x}}_j)}{a}$ so we can use that

$$f'(l) - f'(u) = \mathcal{I}_{\mathbb{R},\mathbb{R}}(l) - \mathcal{I}_{\mathbb{R},\mathbb{R}}(u)$$
$$\leq \mathcal{A}_{L,\mathbb{R}}(\hat{\mathbf{x}}_j - a) - \mathcal{A}_{R,\mathbb{R}}(\hat{\mathbf{x}}_j + a),$$

that $-c, c < \mathcal{A}_{\mathbb{R},\mathbb{R}}(\hat{\mathbf{x}}_j)$, and that

$$\hat{1}_j a^{-1}(f(\hat{\mathbf{x}}_j + a) + f(\hat{\mathbf{x}}_j - a))$$
$$= -\hat{1}_j \left( \frac{f(\hat{\mathbf{x}}_j) - f(\hat{\mathbf{x}}_j - a)}{a} - \frac{f(\hat{\mathbf{x}}_j + a) - f(\hat{\mathbf{x}}_j)}{a} \right) + \hat{1}_j c$$

to produce the final upper bound.

*(Full proof in Appendix A)*

We are now ready to show our main theorem, an upper bound on the number of flips that can be provably robustly classified with a single layer network.

**Theorem 5.8** *(Single-Layer Limit).* No single-layer ReLU-network can provably $\alpha$-robustly classify $\lceil \frac{2}{\alpha} \rceil + 5$ or more flips for any $\alpha \in (0, 1]$.

PROOF OVERVIEW. The proof is by direct application of Lemma 5.7, and expansion of the definition of the derivative. We demonstrate that for the $\hat{\mathbf{x}}$ found by Lemma 5.7, the center of the box must be strictly closer to 0 than its radius. *(Full proof in Appendix A)*

**Corollary 5.9** *(Single-Layer Limit in Multiple Dimensions).* No single-layer ReLU-network can provably $\alpha$-robustly classify $\lceil \frac{2}{\alpha} \rceil + 5$ or more $d$-flips for any $\alpha \in (0, 1]$.

PROOF.    Suppose $f\colon \mathbb{R}^d \to \mathbb{R}$ is a single-layer network that can provably $\alpha$-robustly classify $t \geq \lceil \frac{2}{\alpha} \rceil + 5$ $d$-flips for some $\alpha \in (0,1]$. Then we construct a function $f'\colon \mathbb{R} \to \mathbb{R}$ mapping $x \mapsto f(Mx)$ where $M \in \mathbb{R}^{d \times 1}$ is defined as having $M_{1,1} = 1$ and $M_{i,1} = 0$ for $i > 1$. Then $f'$ can provably $\alpha$-robustly classify $t$ flips for some $\alpha$ and is only 1-layer which contradicts Theorem 5.8. $\qquad\square$

### 5.3   One-layer strong interval-agnostic possibility

***Proposition 5.10*** *(Single-Layer Perfectly Robust Classifiers Always Exist).* For *any* dataset of $n$ points $x_i \in \mathbb{R}$ and labels $l_i \in \{-1, 1\}$ there is a one-hidden-layer ReLU-network that perfectly robustly (but not necessarily provably) classifies it.

PROOF.    We present the construction explicitly. Let $\delta \coloneqq \min\{|x_i - x_j|\colon i \neq j\}$ in

$$f(y) \coloneqq \sum_{i=1}^n l_i \Big( \mathrm{ReLU}\left[ \tfrac{1}{\delta}(y - (x_i - \delta)) \right]$$
$$-\mathrm{ReLU}\left[ \tfrac{2}{\delta}(y - x_i) \right]$$
$$+\mathrm{ReLU}\left[ \tfrac{1}{\delta}(y - (x_i + \delta)) \right] \Big)$$

One can check that this works by plugging in $x_j$, although a full proof is by induction. While this is not immediately of the form described for one-hidden layer networks, one can see easily how to algebraically convert this into that form. Because we only care about robustness, and not interval provability of $f$, this is sufficient. $\qquad\square$

***Proposition 5.11*** *(Perfectly Robust Multi-Dimensional Classifiers Always Exist).* For *any* dataset of $n$ points $x_i \in \mathbb{R}^d$ and labels $l_i \in \{-1, 1\}$ there is a ReLU-network that perfectly robustly (but not necessarily provably) classifies it.

PROOF.    Here we present a special case of the proof shown by Baader et al. (2020), and do not worry about the number of layers or implicit projections involved. Again, we present the construction explicitly. Let $\delta \coloneqq \min\{||x_i - x_j||_\infty \colon i \neq j\}$. We define a neural indicator function $\mathrm{Bump}_k$ for each dimension $k \in [d]$:

$$\mathrm{Bump}_{i,k}(y) \coloneqq \mathrm{ReLU}\left[ \tfrac{1}{\delta}(y_k - (x_{i,k} - \delta)) \right]$$
$$-\mathrm{ReLU}\left[ \tfrac{2}{\delta}(y_k - x_{i,k}) \right]$$
$$+\mathrm{ReLU}\left[ \tfrac{1}{\delta}(y_k - (x_{i,k} + \delta)) \right]$$

We furthermore inductively define a neural implementation of a $d$-value minimum:

$$\mathrm{nmin}_2(a,b) \coloneqq \frac{1}{2}(\mathrm{ReLU}(a+b) - \mathrm{ReLU}(-a-b) - \mathrm{ReLU}(a-b) - \mathrm{ReLU}(-a+b))$$
$$\mathrm{nmin}_d(a_1, \ldots, a_d) \coloneqq \mathrm{nmin}_2(a_1, \mathrm{nmin}_{d-1}(a_2, \ldots, a_d)) \qquad\qquad \text{given } d > 2$$

Finally, we bring this all together in one function:

$$f(y) \coloneqq \sum_{i=1}^n l_i \mathrm{nmin}_d(\mathrm{Bump}_{i,1}(y), \ldots, \mathrm{Bump}_{i,d}(y))$$

We can first see that $\mathrm{nmin}_d(a_1, \ldots, a_d) = min_{i \in [d]} a_i$. Given that $\mathrm{Bump}_{i,k}(y)$ is in $(0, 1]$ if $||y - x_i||_\infty < \delta$ and is $0$ otherwise, we can see that $\mathrm{sign}\, f(y) = \mathrm{sign}\, l_i$ if $||y - x_i||_\infty < \delta$. $\qquad\square$

## 6 Discussion and Future Work

While we limited the scope of our discussion to ReLU-activations, we note that our theorems extend trivially to any monotone bounded activation. However, we observe that non-monotonic activations functions (such as absolute value) do not admit the same forms of theorems. Our preliminary experiments however indicate that substituting ReLU with these activation functions does not result in easier training or provability. This suggests that there are more general versions of the theorems presented here, in particular relating the difficulty of program synthesis with the relational expressiveness of the relaxation used to verify the specification.

## 7 Conclusion

In this work we proved two theorems that demonstrate fundamental limitations on the expressiveness of interval provable neural networks. We showed that no ReLU-network can perfectly provably classify simple one-dimensional datasets containing only three points. This indicates a fundamental loss of precision whenever ReLU-networks are analyzed using interval arithmetic, which can not be regained, no matter the network. Further, we showed that a single hidden layer ReLU-network can not provably classify simple datasets even without the requirement for perfectness, which is in stark contrast to classical universal approximation theorems, where a single hidden layer is sufficient. This shows that the approximate capabilities of interval provable networks are lower compared to standard neural networks.

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

## A  Extended Proofs for Single Hidden Layer Network Results

Here we restate the theorems and show the full proofs for the results in Section 5.2.

**Lemma 5.6** *(End-Neuron Imprecision-Bound).* For all $\kappa \geq 0$ and single-layer ReLU-networks $f$ that classify $k$-flips for $k = \lceil \kappa \rceil + 5$, we have $\kappa < \max\{\mathcal{A}_{L,\mathbb{R}}(\hat{x}_1), \mathcal{A}_{R,\mathbb{R}}(\hat{x}_k)\}$.

PROOF OF 5.6.    We prove this by induction on $c$.

*Induction Hypothesis:*  Given $c \in \mathbb{N} \cup \{0, -1, -2\}$ there is some even natural number $k \leq \mathrm{ReLU}(c+2) + 2$ such that for any single-layer ReLU-network, $f$ that classifies $k$ flips we have

$$c \leq \mathcal{A}_{L,+}(\hat{x}_2) - \mathcal{A}_{R,+}(\hat{x}_2) + \mathcal{A}_{R,+}(\hat{x}_k) - \mathcal{A}_{L,+}(\hat{x}_k),$$
$$\text{and } c \leq \mathcal{A}_{L,-}(\hat{x}_1) - \mathcal{A}_{R,-}(\hat{x}_1) + \mathcal{A}_{R,-}(\hat{x}_{k-1}) - \mathcal{A}_{L,-}(\hat{x}_{k-1}).$$

*Base Case:* Suppose $c \leq 0$.

*Subproof.* Pick $k = 2$. Then

$$\mathcal{A}_{L,+}(\hat{x}_2) - \mathcal{A}_{R,+}(\hat{x}_2) + \mathcal{A}_{R,+}(\hat{x}_2) - \mathcal{A}_{L,+}(\hat{x}_2) = 0 \geq c, \text{ and}$$
$$\mathcal{A}_{L,-}(\hat{x}_1) - \mathcal{A}_{R,-}(\hat{x}_1) + \mathcal{A}_{R,-}(\hat{x}_{2-1}) - \mathcal{A}_{L,-}(\hat{x}_{2-1}) = 0 \geq c. \qquad \triangleleft$$

*Induction Step:* Suppose $c > 0$, and the induction hypothesis holds for $c - 2$.

*Subproof.* Then there is some even natural $k' \leq \mathrm{ReLU}(c-2+2)+2$ such that for any single-layer ReLU-network, $f$, that is a classifier for $k'$ flips we have

$$c - 2 \leq \mathcal{A}_{L,+}(\hat{x}_2) - \mathcal{A}_{R,+}(\hat{x}_2) + \mathcal{A}_{R,+}(\hat{x}_{k'}) - \mathcal{A}_{L,+}(\hat{x}_{k'}),$$
$$\text{and } c - 2 \leq \mathcal{A}_{L,-}(\hat{x}_1) - \mathcal{A}_{R,-}(\hat{x}_1) + \mathcal{A}_{R,-}(\hat{x}_{k'-1}) - \mathcal{A}_{L,-}(\hat{x}_{k'-1}).$$

Pick $k = k' + 2$. Then $k$ is even, and $k \leq \mathrm{ReLU}(c) + 4 \leq \mathrm{ReLU}(c+2) + 2$ since $c > 0$.

Let $f$ be any single-layer ReLU-network that classifies $k$ flips. Then $f$ also classifies $k'$ flips.

We only show the positive bound, that $c \leq \mathcal{A}_{L,+}(\hat{x}_2) - \mathcal{A}_{R,+}(\hat{x}_2) + \mathcal{A}_{R,+}(\hat{x}_k) - \mathcal{A}_{L,+}(\hat{x}_k)$.

The proof for the negative bound is analogous.

There must be some point $l \in [\hat{x}_{k'}, \hat{x}_{k'+1}]$ such that $f'(l) \leq -1$ by the mean value theorem (since ReLU-networks are continuous) and because $f(\hat{x}_{k'}) = 1 = -f(\hat{x}_{k'+1})$.

Similarly, there must be some point $u \in [\hat{x}_{k'+1}, \hat{x}_k]$ such that $1 \leq f'(u)$. Thus,

$$2 \leq f'(u) - f'(l)$$
$$\leq \mathcal{I}_{L,+}(u) + \mathcal{I}_{L,-}(u) + \mathcal{I}_{R,+}(u) + \mathcal{I}_{R,-}(u)$$
$$- \mathcal{I}_{L,+}(l) - \mathcal{I}_{L,-}(l) - \mathcal{I}_{R,+}(l) - \mathcal{I}_{R,-}(l).$$

We know $\mathcal{I}_{L,-}(u) - \mathcal{I}_{L,-}(l) \leq 0$ and $\mathcal{I}_{R,-}(u) - \mathcal{I}_{R,-}(l) \leq 0$ and $\mathcal{I}_{L,+}(t) = -\mathcal{A}_{L,+}(t)$ and $\mathcal{I}_{R,+}(t) = \mathcal{A}_{R,+}(t)$ for any $t$ so

$$2 \leq -\mathcal{A}_{L,+}(u) + \mathcal{A}_{R,+}(u)$$
$$+ \mathcal{A}_{L,+}(l) - \mathcal{A}_{R,+}(l).$$

We also know $\mathcal{A}_{L,S}(t)$ increases as $t$ decreases and $\mathcal{A}_{R,S}(t)$ increases as $t$ increases, so

$$2 \leq -\mathcal{A}_{L,+}(\hat{x}_k) + \mathcal{A}_{R,+}(\hat{x}_k)$$
$$+ \mathcal{A}_{L,+}(\hat{x}_{k'}) - \mathcal{A}_{R,+}(\hat{x}_{k'}).$$

By combining with the positive inductive bound, we get

$$
\begin{aligned}
(c-2)+2 = c \leq\ & \mathcal{A}_{L,+}(\hat{\mathbf{x}}_2) - \mathcal{A}_{R,+}(\hat{\mathbf{x}}_2) + \mathcal{A}_{R,+}(\hat{\mathbf{x}}_{k'}) - \mathcal{A}_{L,+}(\hat{\mathbf{x}}_{k'}) \\
& - \mathcal{A}_{L,+}(\hat{\mathbf{x}}_k) + \mathcal{A}_{R,+}(\hat{\mathbf{x}}_k) + \mathcal{A}_{L,+}(\hat{\mathbf{x}}_{k'}) - \mathcal{A}_{R,+}(\hat{\mathbf{x}}_{k'}) \\
& \leq \mathcal{A}_{L,+}(\hat{\mathbf{x}}_2) - \mathcal{A}_{R,+}(\hat{\mathbf{x}}_2) - \mathcal{A}_{L,+}(\hat{\mathbf{x}}_k) + \mathcal{A}_{R,+}(\hat{\mathbf{x}}_k),
\end{aligned}
$$

which proves the positive bound of the induction hypothesis for $c$. ◁

Thus, by induction, we find that there is some $k \leq \lceil c \rceil + 5$ such that, after removing the negative terms and increasing by swapping $\hat{\mathbf{x}}_2$ with $\hat{\mathbf{x}}_1$ and $\hat{\mathbf{x}}_{k-1}$ with $\hat{\mathbf{x}}_k$:

$$
\begin{aligned}
c + 1 &\leq \mathcal{A}_{L,+}(\hat{\mathbf{x}}_1) + \mathcal{A}_{R,+}(\hat{\mathbf{x}}_k), \\
\text{and}\ \ c + 1 &\leq \mathcal{A}_{L,-}(\hat{\mathbf{x}}_1) + \mathcal{A}_{R,-}(\hat{\mathbf{x}}_k).
\end{aligned}
$$

Summing these equations together gives us:

$$
2c + 2 \leq \mathcal{A}_{L,+}(\hat{\mathbf{x}}_1) + \mathcal{A}_{R,+}(\hat{\mathbf{x}}_k) + \mathcal{A}_{L,-}(\hat{\mathbf{x}}_1) + \mathcal{A}_{R,-}(\hat{\mathbf{x}}_k) = 2\max\{\mathcal{A}_{L,\mathbb{R}}(\hat{\mathbf{x}}_1), \mathcal{A}_{R,\mathbb{R}}(\hat{\mathbf{x}}_k)\},
$$

and thus that $c < \max\{\mathcal{A}_{L,\mathbb{R}}(\hat{\mathbf{x}}_1), \mathcal{A}_{R,\mathbb{R}}(\hat{\mathbf{x}}_k)\}$. □

**Lemma 5.7** (Lower-Bound on Imprecision-Contribution). For any $a \leq 1$ and $k \geq \lceil \frac{2}{a} \rceil + 5$ and single-layer ReLU-network $f$ that classifies $k$ flips, there is some point $j \in [k]$ such that

$$
\begin{aligned}
\hat{1}_j a^{-1} (f(\hat{\mathbf{x}}_j + a) &+ f(\hat{\mathbf{x}}_j - a)) \\
&< \mathcal{A}_{\mathbb{R},\mathbb{R}}(\hat{\mathbf{x}}_j) + \mathcal{A}_{R,\mathbb{R}}(\hat{\mathbf{x}}_j - a) + \mathcal{A}_{L,\mathbb{R}}(\hat{\mathbf{x}}_j + a).
\end{aligned}
$$

PROOF OF 5.7. Let $k \geq \lceil 2a^{-1} \rceil + 5$, and define $c := \frac{2}{a}$. Because $k \geq \lceil |c| \rceil + 5$, we can use Lemma 5.6 to get that $|c| < \max\{\mathcal{A}_{L,\mathbb{R}}(\hat{\mathbf{x}}_1), \mathcal{A}_{R,\mathbb{R}}(\hat{\mathbf{x}}_k)\}$.

For convenience, we define $\tilde{f}_L(x) = a^{-1}(f(x) - f(x - a))$ and $\tilde{f}_R(x) = a^{-1}(f(x + a) - f(x))$.

We only show the proof when $|c| < \mathcal{A}_{L,\mathbb{R}}(\hat{\mathbf{x}}_1)$, the other case is analogous, but picking $j = k$.

In this case we know $0 < \mathcal{A}_{L,\mathbb{R}}(\hat{\mathbf{x}}_1)$ and thus,

$$
\hat{1}_1 a^{-1}(f(\hat{\mathbf{x}}_1 + a) + f(\hat{\mathbf{x}}_1 - a)) \leq \tilde{f}_L(\hat{\mathbf{x}}_1) - \tilde{f}_R(\hat{\mathbf{x}}_1) + c.
$$

There must be a point $l \in [\hat{\mathbf{x}}_1 - a, \hat{\mathbf{x}}_1]$ such that $\tilde{f}_L(l) \leq f'(l)$ and a point $u \in [\hat{\mathbf{x}}_1, \hat{\mathbf{x}}_1 + a]$ such that $f'(u) \leq \tilde{f}_R(u)$.

We can thus derive, in a manner similar to what is seen in Lemma 5.6:

$$
\begin{aligned}
\tilde{f}_L(\hat{\mathbf{x}}_1) - \tilde{f}_R(\hat{\mathbf{x}}_1) \leq f'(l) - f'(u) \leq\ & \mathcal{I}_{L,+}(l) + \mathcal{I}_{L,-}(l) + \mathcal{I}_{R,+}(l) + \mathcal{I}_{R,-}(l) \\
& - \mathcal{I}_{L,+}(u) - \mathcal{I}_{L,-}(u) - \mathcal{I}_{R,+}(u) - \mathcal{I}_{R,-}(u) \\
& \leq \mathcal{I}_{L,-}(l) - \mathcal{I}_{R,-}(u) \\
& \leq \mathcal{A}_{L,\mathbb{R}}(\hat{\mathbf{x}}_1 - a) - \mathcal{A}_{R,\mathbb{R}}(\hat{\mathbf{x}}_1 + a).
\end{aligned}
$$

$\tilde{f}_L(\hat{\mathbf{x}}_1) - \tilde{f}_R(\hat{\mathbf{x}}_1) + c < \mathcal{A}_{\mathbb{R},\mathbb{R}}(\hat{\mathbf{x}}_1) + \mathcal{A}_{L,\mathbb{R}}(\hat{\mathbf{x}}_1 - a) + \mathcal{A}_{R,\mathbb{R}}(\hat{\mathbf{x}}_1 + a)$ as $c \leq |c| < \mathcal{A}_{L,\mathbb{R}}(\hat{\mathbf{x}}_1) \leq \mathcal{A}_{\mathbb{R},\mathbb{R}}(\hat{\mathbf{x}}_1)$. □

**Theorem 5.8** *(Single-Layer Limit).* No single-layer ReLU-network can provably $\alpha$-robustly classify $\lceil\frac{2}{\alpha}\rceil + 5$ or more flips for any $\alpha \in (0, 1]$.

PROOF OF 5.8.     Suppose $\alpha \in (0, 1]$, and assume, for the sake of contradiction, that $f$ is a single-layer ReLU-network with weights $N, M$ and biases $b, d$ that provably $\alpha$-robustly classifies $\lceil\frac{2}{\alpha}\rceil + 5$ flips.

We begin the proof by labeling the intermediate states of interval analysis for a point $\hat{\mathbf{x}}_j$ with interval radius $\alpha$ of the network $f$:

$$v_{\alpha,j}^- := \mathrm{ReLU}(N\hat{\mathbf{x}}_j + b - |N|\alpha)$$
$$v_{\alpha,j}^+ := \mathrm{ReLU}(N\hat{\mathbf{x}}_j + b + |N|\alpha)$$
$$w_{\alpha,j} := v_{j,\alpha}^+ - v_{j,\alpha}^-$$
$$c_{\alpha,j} := v_{j,\alpha}^+ + v_{j,\alpha}^-$$
$$f^\#(\langle\hat{\mathbf{x}}_j, \alpha\rangle) := \langle\frac{1}{2}Mc_{\alpha,j} + d, \frac{1}{2}|M|w_{\alpha,j}\rangle,$$

where the notation $|X|$ means the point-wise absolute value (i.e., $|X|_i = |X_i|$).

Then our assumption for contradiction tells us that for any $j \in [k]$, we have that 0 is not in $f^\#(\langle\hat{\mathbf{x}}_j, \alpha\rangle)$ and thus that $(-1)^j(Mc_{\alpha,j} + 2d) \geq |M|w_{\alpha,j}$.

Let $j$ be such that Lemma 5.7 tells us has $\hat{1}_j\alpha^{-1}(f(\hat{\mathbf{x}}_j+\alpha)+f(\hat{\mathbf{x}}_j-\alpha)) < \mathcal{A}_{\mathbb{R},\mathbb{R}}(\hat{\mathbf{x}}_j)+\mathcal{A}_{R,\mathbb{R}}(\hat{\mathbf{x}}_j-\alpha)+\mathcal{A}_{L,\mathbb{R}}(\hat{\mathbf{x}}_j+\alpha)$.

We note that by expanding the definitions, sums, and meaning of absolute value, we can derive that $Mc_{\alpha,j} + 2d = M(v_{\alpha,j}^+ + v_{\alpha,j}^-) + 2d = f(\hat{\mathbf{x}}_j + \alpha) + f(\hat{\mathbf{x}}_j - \alpha)$. so our assumption for contradiction thus implies $|M|w_{\alpha,j} \leq (-1)^j(Mc_{\alpha,j} + 2d) = (-1)^j(f(\hat{\mathbf{x}}_j + \alpha) + f(\hat{\mathbf{x}}_j - \alpha))$.

We perform the following deduction:

$$
\begin{aligned}
(-1)^j(f(\hat{\mathbf{x}}_j + \alpha) + f(\hat{\mathbf{x}}_j - \alpha)) &\geq \sum_{N_i\hat{\mathbf{x}}_j+b_i\geq-\alpha|N_i|} |M_i|(N_ix + b_i + \alpha|N_i|) - |M|v_{\alpha,j}^- \\
&\geq \sum_{N_i\hat{\mathbf{x}}_j+b_i\geq 0} |M_i|(N_ix + b_i + \alpha|N_i|) - |M|v_{\alpha,j}^- \\
&\geq |M|\mathrm{ReLU}(Nx + b) + \sum_{N_i\hat{\mathbf{x}}_j+b_i\geq 0} \alpha|M_iN_i| \\
&\quad - \sum_{N_i\hat{\mathbf{x}}_j+b_i\geq\alpha|N_i|} |M_i|(N_ix + b_i - \alpha|N_i|) \\
&\geq \sum_{N_i\hat{\mathbf{x}}_j+b_i\geq 0} \alpha|M_iN_i| + \sum_{N_i\hat{\mathbf{x}}_j+b_i\geq\alpha|N_i|} \alpha|M_iN_i| \\
&\geq \alpha\left(\sum_{N_i\hat{\mathbf{x}}_j+b_i\geq 0} |M_iN_i| + \sum_{N_i\hat{\mathbf{x}}_j+b_i\geq\alpha|N_i|} |M_iN_i|\right) \\
&\geq \alpha(\mathcal{A}_{\mathbb{R},\mathbb{R}}(\hat{\mathbf{x}}_j) + \mathcal{A}_{R,\mathbb{R}}(\hat{\mathbf{x}}_j - \alpha) + \mathcal{A}_{L,\mathbb{R}}(\hat{\mathbf{x}}_j + \alpha)) \\
&\geq \hat{1}_j(f(\hat{\mathbf{x}}_j + \alpha) + f(\hat{\mathbf{x}}_j - \alpha))
\end{aligned}
$$

which is a contradiction.                                    $\square$

# B  Extended Proofs for General Impossibility Results

Here we restate the theorems and show the full proofs for the results in Section 4.

## B.1  Proofs for Relative Interior Lemmas

In the following lemmas, let $A, A', B, B', C$ be bounded and non-empty subsets of $\mathbb{R}^d$:

**Lemma 4.3** *(Respects Projection).* $A \sqsubset B$ implies $A|_i \sqsubset B|_i$.

PROOF OF 4.3.   Let $y \in A|_i$. Then there is some $x \in A$ such that $x_i = y$. Then $x \in \mathrm{relint}(B)$ by $A \sqsubset B$. Then there is some $\epsilon > 0$ such that $N_\epsilon(x) \cap \mathrm{aff}(B) \subseteq B$. Then $N_\epsilon(x)|_i \cap \mathrm{aff}(B)|_i \subseteq B|_i$. We know $\mathrm{aff}(B|_i) \subseteq \mathrm{aff}(B)|_i$: given $z \in \mathrm{aff}(B|_i)$, it must be an affine combination of the $i$'th dimension of elements of $B$. Letting $z'$ be the same affine combination of those elements, $z'_i = z$, so $z \in \mathrm{aff}(B)|_i$. Then $N_\epsilon(x|_i) \cap \mathrm{aff}(B|_i) \subseteq B|_i$ and thus $y \in \mathrm{relint}(B|_i)$.  $\square$

**Lemma 4.4** *(Respects Cartesian Product).* $A \sqsubset B$ and $A' \sqsubset B'$ implies $A \times A' \sqsubset B \times B'$.

PROOF OF 4.4.   Let $(x, x') \in A \times A'$. Then because $x \in A$ we know $x \in \mathrm{relint}(B)$ and respectively $x' \in \mathrm{relint}(B')$. Then there is some $\epsilon > 0$ such that $N_\epsilon(x) \cap \mathrm{aff}(B) \subseteq B$, and $\epsilon' > 0$ such that $N_{\epsilon'}(x') \cap \mathrm{aff}(B') \subseteq B'$. We know $(A \cap A') \times (B \cap B') \subseteq (A \times B) \cap (A' \times B')$: $(a, b) \in (A \cap A') \times (B \cap B')$ implies $a \in A \cap A'$ and $b \in B \cap B'$, so $(a, b) \in A \times B$ and $(a, b) \in A' \times B'$ so $(a, b) \in (A \times B) \cap (A' \times B')$.

Then, we know $\mathrm{aff}(B \times B') \subseteq \mathrm{aff}(B) \times \mathrm{aff}(B')$: $(b, b') \in \mathrm{aff}(B \times B')$ implies $(b, b')$ is an affine combination of elements of $B \times B'$ which implies $b$ is an affine combination of elements from $B$ and $b'$ is an affine combination of elements from $B'$, so $(b, b') \in \mathrm{aff}(B) \times \mathrm{aff}(B')$.

Thus for $\lambda = \min\{\epsilon, \epsilon'\}$ we know $N_\lambda(x, x') \cap \mathrm{aff}(B \times B') \subseteq B \times B'$. Thus $(x, x') \in \mathrm{relint}(B \times B')$.  $\square$

**Lemma 4.5** *(Downward Union).* $A \sqsubset C$ and $B \sqsubset C$ implies $A \cup B \sqsubset C$.

PROOF OF 4.5.   Suppose $x \in A \cup B$. Then $x \in A$ or $x \in B$. Either way, we know $x \in \mathrm{relint}(C)$.  $\square$

**Lemma 4.6** *(Downward Hull).* $C \in \mathscr{B}^d$ and $A \sqsubset C$ implies $\mathcal{H}_\infty(A) \sqsubset C$.

PROOF OF 4.6.   $C \in \mathscr{B}^d$ implies $C|_i \in \mathscr{B}$, and thus $C|_i$ is convex so $\mathrm{relint}(C|_i)$ is convex. We know $A|_i \sqsubset C|_i$ by Lemma 4.3, so $\mathcal{H}_\infty(A|_i) \sqsubset C|_i$ by convexity of $C|_i$ and that $\mathcal{H}_\infty$ is the convex hull in one dimension. Thus, by Lemma 4.4, we know $\mathcal{H}_\infty(A|_1) \times \cdots \times \mathcal{H}_\infty(A|_d) \sqsubset C|_1 \times \cdots \times C|_d$. Because $C \in \mathscr{B}^d$ we know $C = C|_1 \times \cdots \times C|_d$ and similarly that $\mathcal{H}_\infty(A) = \mathcal{H}_\infty(A|_1) \times \cdots \times \mathcal{H}_\infty(A|_d)$. Thus, $\mathcal{H}_\infty(A) \sqsubset C$.  $\square$

## B.2    Proofs for Inversion and Impossibility Theorems

**Lemma 4.9** *(Concrete Relative Inversion).* Suppose $f$ is a feed-forward network with ReLU-activations and $Y, X' \in \mathscr{B}^d$ and $X$ is compact and non-empty. Then
$$Y \sqsubset \mathcal{H}_\infty(f(X)) \implies \exists X' \sqsubset \mathcal{H}_\infty(X). Y \subseteq f^\#(X').$$

PROOF OF 4.9.    The proof is by structural induction on the construction of the network $f$, assuming the lemma itself as the induction hypothesis for any network with fewer operations than $f$. First, assume $Y \sqsubset \mathcal{H}_\infty(f(X))$.

*Case: $f = g \circ h$.*                                                                          *(Sequential Computation)*

*Subproof.* By definition, $Y \sqsubset \mathcal{H}_\infty(g \circ h(X))$. Thus, there exists some $H \sqsubset \mathcal{H}_\infty(h(X))$ such that $Y \subseteq g^\#(H)$ by the induction hypothesis on $g$. Applying the induction hypothesis again with the network $h$, we get a set $X' \sqsubset \mathcal{H}_\infty(X)$ such that $H \subseteq h^\#(X')$. Thus, $Y \subseteq g^\#(H) \subseteq g^\# \circ h^\#(X') = f^\#(X')$.    ◁

*Case: $f = \mathsf{Dup}$.*                                                                                      *(Duplication)*

*Subproof.* Then $Y|_1 \sqsubset \mathcal{H}_\infty(X)$ and $Y|_2 \sqsubset \mathcal{H}_\infty(X)$ by Lemma 4.3, We choose $X' = \mathcal{H}_\infty(Y|_1 \cup Y|_2)$ which we know by Lemma 4.5 and Lemma 4.6, is such that $X' \sqsubset \mathcal{H}_\infty(X)$. Thus, $Y|_1 \subseteq X'$ and $Y|_2 \subseteq X'$, so $Y \subseteq X' \times X' = f^\#(X')$.    ◁

*Case: $f = \langle g_1, g_2 \rangle$.*                                                                                  *(Parallel)*

*Subproof.* Then $Y|_1 \sqsubset \mathcal{H}_\infty(g_1(X|_1))$ and $Y|_2 \sqsubset \mathcal{H}_\infty(g_2(X|_2))$ by definition. Then applying the induction hypothesis twice produces $L \sqsubset \mathcal{H}_\infty(X|_1)$ and $R \sqsubset \mathcal{H}_\infty(X|_2)$ such that $Y|_1 \subseteq g_1^\#(L)$ and $Y|_2 \subseteq g_2^\#(R)$. Then we choose $X' = L \times R$ which we know by Lemma 4.4, is such that $X' \sqsubset \mathcal{H}_\infty(X)$. Then $Y \subseteq g_1^\#(X'|_1) \times g_2^\#(X'|_2) = f^\#(X')$.    ◁

*Case: $f = c$.*                                                                                          *(Constant)*

*Subproof.* Here, any subset $X' \sqsubset X$ will suffice. Then we can let $X' = \{\mathcal{C}(X)\}$.    ◁

*Case: $f = c\cdot$ for $c \neq 0$.*                                                          *(Multiplication by a Constant)*

*Subproof.* Let $X' = \mathcal{B}_{|c^{-1}|\mathcal{R}(Y)}(c^{-1}\mathcal{C}(Y))$. Then clearly, $Y \subseteq f(X')$. It remains to show that $X' \sqsubset \mathcal{H}_\infty(X)$. For the remainder of this subproof, because we know that $d = 1$ we will write $x_l = \inf X$, $x_u = \sup X$, $y_l = \inf Y$ and $y_u = \sup Y$. We note that $x_l = \mathcal{C}(\mathcal{H}_\infty(X)) - \mathcal{R}(\mathcal{H}_\infty(X))$ and so on. Supposing $c > 0$ (the other case is analogous) and $x_l < x_u$ (the proof is similar when they are equal), we have $cx_l < y_l \leq y_u < cx_u$ by $Y \sqsubset \mathcal{H}_\infty(f(X))$.

Then we know $x_l < |c^{-1}|y_l \leq |c^{-1}|y_u < x_u$, and thus $X' \sqsubset \mathcal{H}_\infty(X)$.    ◁

*Case: $f = \mathsf{ReLU}$.*                                                                                      *(Activation)*

*Subproof.* Again, because we know that $d = 1$ we will write $x_l := \inf X$, $x_u := \sup X$, and $y_l := \inf Y$ and $y_u := \sup Y$. We know that $\inf \mathcal{H}_\infty(f(X)) = \text{ReLU}(x_l)$ and $\sup \mathcal{H}_\infty(f(X)) = \text{ReLU}(x_u)$. Thus by $Y \sqsubset \mathcal{H}_\infty(f(X))$ we know $\text{ReLU}(x_l) \leq y_l \leq y_u \leq \text{ReLU}(x_u)$. We then have two cases we need to address:

*Suppose: $x_u > 0$.*

*Subproof.* Here we define $X' = [y_l, y_u]$. We thus have $x_l \leq \text{ReLU}(x_l) < y_l \leq y_u < \text{ReLU}(x_u) = x_u$ provided $x_l < x_u$. Otherwise we know $x_l = \text{ReLU}(x_l) = y_l = y_u = \text{ReLU}(x_u) = x_u$ so we have $X' \sqsubset \mathcal{H}_\infty(X)$. ◁

*Suppose: $x_u \leq 0$.*

*Subproof.* Define $X' = \{\frac{x_u + x_l}{2}\}$. $\text{ReLU}(\frac{x_u + x_l}{2}) = 0 = y_l = y_u$ implies $X' \sqsubset \mathcal{H}_\infty(X)$. ◁

Thus, in both cases we can find $X' \sqsubset \mathcal{H}_\infty(X)$ such that $Y \subseteq f(X') \subseteq f^{\#}(X')$ ◁

*Case: $f = \mathsf{Sum}$.* *(Addition)*

*Subproof.* Conveniently again, $Y$ is one-dimensional. Either $\mathcal{H}_\infty(f(X))$ is a single point or it is not:

*Assume: $\mathcal{H}_\infty(f(X))$ is a single point.*

*Subproof.* Then $\inf Y = \sup Y = \inf \mathcal{H}_\infty(f(X)) = \sup \mathcal{H}_\infty(f(X))$. Let $y = \inf Y$. In this case, we know there is some compact and non-empty set $Z \subseteq \mathbb{R}$ such that $X = \{(x', y-x') : x' \in Z\}$. Then we can pick $X' = \{(\mathcal{C}(\mathcal{H}_\infty(Z)), y - \mathcal{C}(\mathcal{H}_\infty(Z)))\}$ which is the singleton-set containing the center of the $l_\infty$-hull of $X$ and thus $X' \sqsubset \mathcal{H}_\infty(X)$ by Lemma 4.8. ◁

*Otherwise: $\mathcal{H}_\infty(f(X))$ is not a single point.*

*Subproof.* We know $\inf \mathcal{H}_\infty(f(X)) < \inf Y \leq \sup Y < \sup \mathcal{H}_\infty(f(X))$. Because $Y$ is one-dimensional and a relative subset of the non-singular $\mathcal{H}_\infty(f(X))$ we know $Y \sqsubset \mathcal{H}_\infty(f(\mathcal{H}_\infty(X)))$.

Let $a, b, r_a, r_b, y, r_y$ be as follows:

$$a = \mathcal{C}(\mathcal{H}_\infty(X|_1)), \qquad r_a = \mathcal{R}(\mathcal{H}_\infty(X|_1)),$$
$$b = \mathcal{C}(\mathcal{H}_\infty(X|_2)), \qquad r_b = \mathcal{R}(\mathcal{H}_\infty(X|_2)),$$
$$y = \mathcal{C}(Y), \qquad \text{and } r_y = \mathcal{R}(Y).$$

Then $\mathcal{H}_\infty(X) = \mathcal{B}_{r_a}(a) \times \mathcal{B}_{r_b}(b)$ and thus $Y \sqsubset f(\mathcal{B}_{r_a}(a) \times \mathcal{B}_{r_b}(b)) = \mathcal{B}_{(r_a + r_b)}(a + b)$.

Choose $X' = \mathcal{B}_r(x)$ for $x$ and $r$ defined as:

$$x_1 = a + r_a \frac{y - a - b}{r_a + r_b}, \qquad r_1 = \frac{r_y r_a}{r_a + r_b},$$
$$x_2 = b + r_b \frac{y - a - b}{r_a + r_b}, \quad \text{and } r_2 = \frac{r_y r_b}{r_a + r_b}.$$

Then clearly, $x_1 + x_2 = y$, and $r_1 + r_2 = r_y$ so $Y \subseteq f(X')$.

Thus $r_y < r_a + r_b$ by $Y \sqsubset B_{r_a + r_b}(a + b)$.

This also tells us that $y + r_y < a + b + r_a + r_b$ and $y - r_y > a + b - r_a - r_b$. If $r_a \neq 0$ we can derive $x_1 + r_1 < a + r_a$ and $x_1 - r_1 > a - r_a$. Similarly, if $r_b \neq 0$ we can derive $x_2 + r_2 < b + r_b$ and $x_2 - r_2 > b - r_b$. Thus, $X_1 \sqsubset \mathcal{B}_{r_a}(a)$ and $X_2 \sqsubset \mathcal{B}_{r_b}(b)$, and thus by Lemma 4.4 we have $X' \sqsubset \mathcal{B}_{r_a}(a) \times \mathcal{B}_{r_b}(b) = \mathcal{H}_\infty(X)$. ◁

Thus, in both cases, there exists an $X' \sqsubset \mathcal{H}_\infty(X)$ such that $Y \subseteq f(X') \subseteq f^\#(X')$. ◁

As any feed forward neural network (without input-value dependent loops) can be expressed using these operations without modifying the result under interval analysis, by induction $\exists X' \sqsubset \mathcal{H}_\infty(X).\, Y \subseteq f^\#(X')$. □

**Theorem 4.10** (*Fundamental Imprecision of Interval*). Suppose $f\colon S \to T$ with $S, T \in \mathscr{T}$ is a ReLU-network and $y \in \mathbb{R}^m$ and $N \subseteq f^{-1}(y)$ is compact and non-empty. Then assuming $M \in \mathscr{B}^S$:
$$\exists M \sqsubset \mathcal{H}_\infty(N). \, y \in f^\#(M).$$

PROOF OF 4.10.    The proof is by structural induction on the construction of the network $f$, assuming the theorem itself as the induction hypothesis for any network with fewer operations than $f$.

Let $f\colon \mathbb{R}^n \to \mathbb{R}^m$ be a feed forward network with ReLU activations, and let $y \in \mathbb{R}^m$ and let $N \subseteq f^{-1}(y)$ be compact and non-empty. Then $f$ is one of the following cases:

*Case:* $f = g \circ h$                                                                                   *(Sequential Computation)*

*Subproof.* We first know that $N \subseteq h^{-1} \circ g^{-1}(y)$ by the definition of $f$. We then infer that $h(N) \subseteq \mathcal{H}_\infty(N)$ is compact and non-empty by application of the continuous function $h$. Thus, by induction on $h(N)$ and $g$ there is some $M' \sqsubset \mathcal{H}_\infty(h(N))$ such that $y \in g^\#(M')$. Thus, by Lemma 4.9, we know that there is some $M \sqsubset \mathcal{H}_\infty(N)$ such that $M' \subseteq h^\#(M)$. Thus, $y \in g^\# \circ h^\#(M) = f^\#(M)$.                   $\triangleleft$

*Case:* $f = \mathsf{Dup}$.                                                                                              *(Duplication)*

*Subproof.* We have $N \subseteq \{y_1\} \cap \{y_2\}$ so $N = \{y_1\}$. By singleton reflexivity, $N \sqsubset \mathcal{H}_\infty(N)$. Thus, $y \in f^\#(N)$.
                                                                                                                         $\triangleleft$

*Case:* $f = \langle g_1, g_2 \rangle$.                                                                                            *(Parallel)*

*Subproof.* First we know $N|_1 \subseteq g_1^{-1}(y_1)$ and $N|_2 \subseteq g_2^{-1}(y_2)$ by projection and that $N|_1$ and $N|_2$ are still compact and non-empty. Thus, by the induction hypothesis twice we see that there are boxes $M_1 \sqsubset \mathcal{H}_\infty(N|_1)$ and $M_2 \sqsubset \mathcal{H}_\infty(N|_2)$ such that $y_1 \in g_1^\#(M_1)$ and $y_2 \in g_2^\#(M_2)$. Then $M_1 \times M_2 \sqsubset \mathcal{H}_\infty(N)$ by Lemma 4.4. Then $y_1 \in f^\#(M_1 \times M_2)|_1$ and $y_2 \in f^\#(M_1 \times M_2)|_2$ by soundness. Thus, there is some box $M \sqsubset \mathcal{H}_\infty(N)$ such that $y \in f^\#(M)$.                   $\triangleleft$

*Case:* $f = c$.                                                                                                 *(Constant)*

*Subproof.* We know $y = c$ and thus $f^{-1} = \mathbb{R}$. If we let $M = \{\mathcal{C}(\mathcal{H}_\infty(N))\} \sqsubset \mathcal{H}_\infty(N)$, then $y \in f^\#(M)$.    $\triangleleft$

*Case:* $f(y) = c \cdot y$ for $c \neq 0$.                                                            *(Multiplication by a Constant)*

*Subproof.* We know $f^{-1}(y) = \{c^{-1} \cdot y\} = N \sqsubset \mathcal{H}_\infty(N)$ by $N$ being non-empty and thus $y \in f^\#(N)$.                   $\triangleleft$

*Case:* $f = \mathrm{ReLU}$.                                                                                       *(Activation)*

*Subproof.* Then $y = \mathrm{ReLU}(x)$ can either be zero or greater than zero.

> *Case:* $y > 0$
>
> *Subproof.* $N = \{y\}$ by def. of ReLU, and we know $\{y\} \sqsubset \mathcal{H}_\infty(N)$ and $y \in f^\#(\{y\})$.                   $\triangleleft$

*Case:* $y = 0$

*Subproof.* $N = (-\infty, 0]$ by def. of ReLU. Thus $\{\mathcal{C}(\mathcal{H}_\infty(N))\} \sqsubset \mathcal{H}_\infty(N)$ and $y \in f^\#(\{\mathcal{C}(\mathcal{H}_\infty(N))\})$.

$\triangleleft$

Because $y$ is the result of a ReLU, it must have been one of these two possibilities, and in both cases we could find some $M \sqsubset \mathcal{H}_\infty(N)$ such that $y \in f^\#(M)$.

$\triangleleft$

*Case:* $f = \mathsf{Sum}.$ *(Addition)*

*Subproof.* In this case, we know $N \subseteq f^{-1}(y) = \{(a, y - a) \colon a \in \mathbb{R}\}$. We pick $M = \{\mathcal{C}(\mathcal{H}_\infty(N))\} \sqsubset \mathcal{H}_\infty(N)$. Given $N$ is bounded, we know:

$$\mathcal{C}(\mathcal{H}_\infty(N)) = \left( \frac{\inf N|_1 + \sup N|_1}{2}, \frac{\inf N|_2 + \sup N|_2}{2} \right)$$
$$= \left( \frac{\inf N|_1 + \sup N|_1}{2}, \frac{2y - \inf N|_1 - \sup N|_1}{2} \right).$$

We can rewrite $f(\mathcal{C}(\mathcal{H}_\infty(N)))$ as

$$f(\mathcal{C}(\mathcal{H}_\infty(N))) = \frac{\inf N|_1 + \sup N|_1}{2} + \frac{2y - \inf N|_1 - \sup N|_1}{2} = y.$$

Thus, $y = f(\mathcal{C}(\mathcal{H}_\infty(N))) \in f^\#(M)$

$\triangleleft$

Thus, $\exists M \sqsubset \mathcal{H}_\infty(N).\, y \in f^\#(M)$.

$\square$

## C  Reference of Symbols and Notation

### General Notation (Sec. 3.1)

$[k]$    The *index set* $\{1, \dots, k\}$ for $k \in \mathbb{N}$

$Y|_i$    The *restriction* of $Y \subseteq T_1 \times \dots \times T_k$ to dimension $i$, or formally, the set $\{y_i \colon y \in Y\}$

$\mathcal{P}$    The *powerset operator*

$f[S]$    The lifted application of $f$ to a subset $S$ of its domain. Formally, the set $\{f(x) \colon x \in S\}$

$S^\circ$    The *topological interior* of $S$

ReLU    The function mapping $x \in \mathbb{R}$ to $\max\{x, 0\}$

### Spaces and Domains (Sec. 3.1)

$\mathbb{N}$    The set of *natural numbers*

$\mathbb{R}$    The set of *real numbers*

$\mathscr{T}$    The class of *binary-tree tensor-spaces* defined as $\{T_1 \times T_2 \colon T_1, T_2 \in \mathscr{T}\} \cup \{\mathbb{R}\}$

$\mathscr{B}^S$    The set of closed, non-empty, axis-aligned *boxes* over $S \in \mathscr{T}$

$\mathscr{B}^d$    For $d \in \mathbb{N}$, this is the set of closed, non-empty, axis-aligned *boxes* over $\mathbb{R}^d$

### Notation for Boxes (Sec. 3.1)

$\mathcal{B}_r(c)$    The *closed box* with center $c \in \mathbb{R}^d$ and radius $r \in \mathbb{R}^d_{\geq 0}$

$\mathcal{R}(B)$    The *radius* of the box $B$

$\mathcal{C}(B)$    The *center* of the box $B$

$\mathcal{H}_\infty(S)$    The $l_\infty$-*hull*, or smallest axis aligned box containing $S$

### Notation for Analysis (Sec. 3.2)

$f^{\mathcal{P}} \colon \mathcal{P}(S) \to \mathcal{P}(T)$    The *perfect transformer*, defined as the mapping $S \mapsto \{f(x) \colon x \in S\}$

$f^{\#} \colon \mathscr{B}^S \to \mathscr{B}^T$    The *interval transformation* of a $\sigma$-network $f \colon S \to T$

### Relative Subsets (Sec. 4.1)

$\mathrm{aff}(S)$    The smallest *linear-subspace* of $\mathbb{R}^d$ containing $S$

$\mathrm{relint}(S)$    The *relative interior*. Formally, the set $\{x \in S \colon \exists \epsilon > 0.\, \mathcal{B}^\circ_\epsilon(x) \cap \mathrm{aff}(S) \subseteq S\}$

$A \sqsubset B$    The *relative subset* relation, defined formally, as $A \subseteq \mathrm{relint}(B)$

### Single-Layer Networks (Sec. 5.2)

$\hat{\mathbf{x}}_i$    An input point in $\mathbb{R}^d$ defined as $\hat{\mathbf{x}}_{i,1} \coloneqq 2i$ and $\hat{\mathbf{x}}_{i,j} = 0$ for $j > 1$

$\hat{\mathbf{1}}_i$    An output label equal to $(-1)^i$

$\mathcal{A}_{D,S}(x)$    The absolute *imprecision-contribution*

$\mathcal{I}_{D,S}(x)$    The standard *imprecision-contribution*

