# OpenReview forum: "The Fundamental Limits of Neural Networks for Interval Certified Robustness"
_TMLR — Accepted by TMLR_

### Review · Reviewer_N9Hb · 2022-06-20

**Summary Of Contributions:**

The paper attempts to investigate the class of neural networks for which precise interval analysis/interval bound propagation, for verifying robustness of deep neural networks, is feasible. A negative result is established.

**Broader Impact Concerns:**

There are no broader impact concers.

**Requested Changes:**

The proofs can be made more tractable by introducing a symbol table earlier in the paper.

**Strengths And Weaknesses:**

A negative result is presented that non-invertible functions can’t be built using affine transformations and ReLUs such that over-approximation using interval analysis is precise everywhere. Further, the paper claims that even when the requirement for perfect provability is relaxed to regions that are distant from each other (\alpha-interval provable with < 1), there are datasets with O(1/alpha) points that can not be provably robustly classified with one-hidden layer networks using interval analysis. Finally, the paper establishes that perfectly robust classifiers can always be constructed with just one-hidden layer networks, even if they are not necessarily provably robust using interval analysis.

The reviewer has not yet gone over the proofs in detail, but the overall results appear to be consistent.

Typos and minor comments:
Abstract has repeated "there are" in the last sentence.

---

> ### Author Response · Authors · 2022-07-14
> **Added symbol table**
>
> > The proofs can be made more tractable by introducing a symbol table earlier in the paper.
>
> We think this is a great idea.  However, to accommodate the full collected set of symbols and definitions, we have given this its own full-page at the end of the paper in Appendix C.

---

### Review · Reviewer_ubCf · 2022-06-23

**Summary Of Contributions:**

Analysis of neural networks for robustness has become an active research topic. This paper argues that even if robust classifiers exist, they may not be provable by interval propagation. Interval propagation is a scalable but imprecise method to abstractly analyse programs and neural networks. It is therefore not surprising that there exist robust classifiers that cannot be proved to be robust using interval propagation. The paper identifies a class of neural networks (non-invertible networks with ReLU activations) interval propagation loses precision on and shows a concrete example of a dataset for which an interval-verifiable robust classifier cannot be constructed. It considers a less strict definition of robustness ($\alpha$-robust) and constructs another example dataset (called flips) for which a single layer network verifiable with intervals is not possible. These datasets do admit robust classifiers.

**Requested Changes:**

The paper needs a significant rewrite to address the points raised above.

Do not use contractions like “can’t” in formal writing.

Use \citep for citing related work.


**Strengths And Weaknesses:**

Strengths
---

Knowing limitations of verifiable neural networks is an interesting problem. This paper reports on deficiency of interval propagation as a means of verification.

Weaknesses
---

The definition of “relative” interior is central to the paper, but I could not get an intuition about the significance of it. What is a “non-relative” interior and what difficulty does it pose?

The writing is sloppy and has many mistakes.

* Sect. 3.2: The definition of $r$ is missing.
* What do $m$ and $n$ refer to in Def. 3.2?
* How can multiplication by a constant in Def. 3.2 be a singleton set?
* In Def. 4.1, what is $\mathbf{B}_\epsilon^o$?
* Def. 4.1 seems to admit points lying on the hyper-planes of a box (so that $aff(S) = S$). But isn’t this supposed to define an "interior"? Not clear if the paragraph following the definition is meant to take care of this case.
* The last sentence in the paragraph after Def. 4.2 has many mistakes.
* Lemma 4.6: What is $H_\infty$?
* Def. 5.1: Why does $i$ range over the dimensions?

The claim below is stated in more general terms than demonstrated in the paper.

“Given this, we derive a paradox: while every dataset can be robustly classified, there are there are (sic) simple datasets that can’t be provably robustly classified with interval analysis.” – The term “every dataset” is misleading. The claim (Proposition 5.9) is stated for 1D datasets.

I did not understand the following statements.
1. We note that while Definition 3.2 defines an ordering of addition, the above definition does not.
2. This suggests that there are more general versions of the theorems presented here, in particular relating the difficulty of program synthesis with the relational expressiveness of the relaxation used to verify the specification.

The authors need to proofread the paper to fix grammatical and typographic mistakes.

---

> ### Author Response · Authors · 2022-07-04
> **Clarifications and Improvements**
>
> ## The definition of “relative” interior is central to the paper, but I could not get an intuition about the significance of it. What is a “non-relative” interior and what difficulty does it pose?
>
> Figure 2 visualizes the concept of a relative interior.  In 2b, the green box is a subset of the “non-relative” interior, or standard topological notion of interior of the purple box. Formally, the topological interior of a set $X$ in point set topology is defined to be all those points which are members of an open neighborhood contained by $X$.  Here the green box doesn’t touch any of the “walls” of the purple box, and thus if we were to “remove the walls” the green box would still be its subset.  In 2c however, the green box touches the surface of the purple line, but since the green and purple lines have no volume, or are lower dimensional than the space needed to represent them, we still want to be able to say that the green line is contained within the “interior that matters” of the purple line.  The “interior that matters” here is the purple line not including its end-points.  In this case however, the traditional, non-relative interior would be the empty-set.
>
> ## The writing has many mistakes…
>
> We thank the reviewer for finding these and have have fixed these in the revision.  Some of the points here which are in fact not mistakes we clarify below.
>
> **Sect. 3.2: The definition of $r$ is missing.**
>
> This is the “radius” of the interval $B$ defined in the second sentence of the “interval analysis” paragraph.
>
> **In Def. 4.1, what is $\mathcal{B}_\epsilon^\circ(x)$?**
>
> This refers to the standard topological interior of the $\epsilon$-sized $l_\infty$-ball around $x$.  It is defined in section 3.1.
>
>
> **Def. 4.1 seems to admit points lying on the hyper-planes of a box**
>
> This is a well known and established definition of relative interior which can be found on [wikipedia](https://en.wikipedia.org/wiki/Relative_interior). The definition does not imply that $relint(S) = S$.  For a d-dimensional $l_\infty$-ball, $B$ embedded in d-dimensions, this is in fact equivalent to the standard topological definition of interior.   One can see this by plugging in a surface point $x$. In this case, $aff(B) = \mathbb{R}^d$ so any open neighborhood $N$ of $x$ is a subset of $aff(B)$.  Thus, $N$ is not a subset of $B$ since $x$ is a surface point.
>
> **Lemma 4.6: What is $H_\infty$?**
>
> This is the $l_\infty$-hull defined in section 3.1.
>
>
> ## The claim below is stated in more general terms than demonstrated in the paper: “Given this, we derive a paradox: while every dataset can be robustly classified, there are there are (sic) simple datasets that can’t be provably robustly classified with interval analysis.” – The term “every dataset” is misleading. The claim (Proposition 5.9) is stated for 1D datasets.
>
> We have fixed this by adding Proposition 5.11 in the updated version.
>
>
> ## I did not understand the following statement: We note that while Definition 3.2 defines an ordering of addition, the above definition does not.
>
> As this is only meaningful for domains beyond interval, we have removed it.
>
>
> ## I did not understand the following statement:  This suggests that there are more general versions of the theorems presented here, in particular relating the difficulty of program synthesis with the relational expressiveness of the relaxation used to verify the specification.
>
> In abstract interpretation, the incomplete analysis is frequently made more efficient by altering the kinds and number of relationships between program variables that the domain can express.  The most precise convex domain, polyhedra, can represent arbitrary linear relationships between program variables (or neuron outputs).  The interval domain on the other hand can not represent any relationships between variables or neurons.  A relational domain like polyhedra can express for example that $0 < f(x)_1 = f(x)_2 < 10$ for any $x$.  On the other hand, intervals can only express $0 < f(x)_1 < 10$ and  $0 < f(x)_2 < 10$.  The fact that adding a transformer which accounts for some “relationality” sidesteps the theorems presented in the paper but does not immediately improve training implies a generalized theory.  Specifically, this likely means that there is a generalized version of our theorems that accounts for increases in the expressiveness, or “relationality”, of the analysis domain.

---

> > ### Comment · Reviewer_ubCf · 2022-07-13
> > **Looked at the revision**
> >
> > Thank you for your response and updates to the paper. Thm. 5.11 is now added to handle a multi-dimensional case.
> >
> > Some more suggestions:
> > * Definition of $r$ is missing: Shouldn't the LHS of the last line of the Interval Analysis paragraph be $r$?
> > * Multiplication by a constant: Shouldn't it be $\{\kappa \cdot x : x \in B$?
> > * Def. 4.1: Because this is a standard definition, you should reword it as "We recall the standard definition of ..." to avoid confusion.

---

> > > ### Author Response · Authors · 2022-07-13
> > > **Thanks for the further comments**
> > >
> > > We have corrected these and followed the suggestions in the revision.

---

### Review · Reviewer_f5Ug · 2022-06-28

**Summary Of Contributions:**

In this paper authors investigate robustness (adversarial robustness / math context of stable solution in well-posed problems) on interval neural networks. The main contributions of the paper:
- theorem and its proof "Fundamental imprecision of intervals": for the feed forward network with ReLU activation $f: \mathbb{R}^n\to\mathbb{R}^m$ under some conditions there could exist some input interval where the network produces approximation which includes values that are not possible / true / concrete network outputs. This means that interval network looses precision.
- based on this theorem and introducing definition of provably robust classifier authors show simple 1d binary classification dataset with 3 points for which there is no feed forward ReLU network which is perfectly 1-provably robust.
- if we relax definition to have $\alpha$-interval provable robust classifier, authors showed that datasets with $O(\alpha^{-1})$ points from $\mathbb{R}$ (binary classification task) cannot be $\alpha$-provable robust classified by a single-layer ReLU network (definition of this kind of network see in def 5.4)
- finally authors demonstrate paradox by showing that any dataset with points from $\mathbb{R}$ can be perfectly robustly classified by a single-layer ReLU network while from above it is impossible for interval case.

**Broader Impact Concerns:**

The paper has analysis of robustness for interval networks. The whole paper is abstract math analysis and abstract examples. One suggestion maybe is to have one-two sentences why robustness is important and what is the practical relation could be of the theory in the paper for those who is using interval networks in practice.

**Requested Changes:**

General:
- I never saw notation for "for $\forall x.f(x)>0$" with dot between expressions. For me personally it is very annoying to read. Maybe use comma, or space or $\Rightarrow$

Abstract
- "there are there are" -> "there are"

Figure 2
- Add transparency for squares to make visible intersection between squares of different color. Or use different hatching for them without fill. Without this it is very unclear for "after the 2nd affine layer ..." statement in the caption.
- Specify axis for (c)-(h) pictures, that x axis correspond to the upper path in (a) and y axis corresponds to lower path in (a). Maybe it is good to show how input interval to the NN is mapped into 2D plane first to make it clear.

Sec. 3.1
- I propose to remove notation $\mathbb{B}$ as it messes things with $\mathbb{R}$ and simply use $B$, in context either we use $B$ or $B_r(c)$. I don't think there should be any differences in notation here.
- definition of $H_\inf(C)$ seems messed sup and inf, should be changed in left and right inequalities.
- Move $[k]:=\\{1,..k\\}$ notation to the beginning.

Sec 3.2
- first paragraph: $\mathbb{B}_\epsilon(x)$ -> $\mathbb{B}^0_\epsilon(x)$. Even I think everywhere we can skip $0$ in the superscription as it is not used in any other context.
- second paragraph second sentence - it is needed to rephrase it as very hard to read and follow.
- Maybe I missed, but I think Def 3.1 and 3.2 are given only for 1d input in the end, it cannot give us definition on multi-dimensional inputs. The way we can get multi-dim is duplication, which doesn't create different coordinates. With this I think all proofs and theorems / results cannot be extended to the most interesting case of multi-dimensional inputs.
- Usage of $x_1$, $x_2$ is not clear and the whole notation is messed between 1d and nd cases. So it should be clarified, e.g. if $x$ is a vector or not, from which space it is.
- Def 3.2 Parallel: maybe add that $B=(x_1, x_2)$?
- Def 3.2 Constant: what is $m$, $n$? what is the notation for $\\{k\\}$
- Proposition 3.3 - what does it mean over-approximate? maybe have footnote or definition on that?

Sec. 4
- Harder to read when notation on $f[B]$ and $f(B)$ is used in the same sentence but no any reason why to mix it.

Sec. 4.1
- Are notes for both definitions 4.1 and 4.2 given in case of one box or in case $S, A, B$ are set of boxes? It is not clear from the text. For the def 4.1 note why do we need to have center point in the relint$(S)$? Also here $x\in S|^0_i$ should be $x_i\in S|^0_i$. In note of def 4.2 seems a lot of typos in the $[a,b]$, $[b,c]$ and $a'$, $b'$ usage.
- last paragraph - could authors add any examples for both cases?

Sec 4.2
- In Lemma 4.9 "By Lemma 4.5" -> "By Lemma 4.5 and 4.6" (same in Appendix)
- In Lemma 4.9 $f(x_1, x_2) = (g_1(x_1), g_2(x_2))$ case: $Y|_2$ should be relative interior of $H_\inf(g_2(x_2))$, samy $Y|_2$ should be subset of $g^\\#_2(R)$. Same fixes are needed in Appendix
- Lemma 4.9 no any proof for multi-dimensional inputs and results cannot be simply extended to multi-dimensional case.

Sec 4.3
- I found it is hard to read with changed notation on $x$ where it is not input anymore and now used in the context of function value with $f^-1(x)$.  This also creates difficulties to read Fig. 3 as took time to understand what is x-axis and what is y-axis. Also notation on points introduced later - also hard to parse earlier.
- th. 4.10: "using the lemma itself" -> "using the theorem itself"
- th. 4.10: function takes $\mathbb{R}^n$ but in proof there is no any justification for this, only for $\mathbb{R}$
- th. 4.10: case $f(x_1, x_2) = (g_1(x_1), g_2(x_2))$: fix notation as we swtiched to $y$ here as input space. Same in Appendix. Also "$x_1\in g_1^\\#(M_1\times M_2)$ and $x_2\in g_2^\\#(M_1\times M_2)$" should be $x\in g_1^\\#(M_1) \times g_2^\\#(M_2)=f^\\#(M)$
- I like here example of fig. 3 to demonstrate the outcome from the theorem, but it is not clear how to use this theorem in general, and why it is about imprecision of interval? Maybe some extended comment / clarification, or even abstract explanation in case $N$ is some set of the points which is mapped to the same $x$ (so non-invertible on this point).

Sec. 5:
- Corollary 5.2 what about other $\nu$?
- Def. 5.3 for $F$ is given not clearly.
- the whole section seems weak to me as results only for 1d input which is not much interesting. Can we inherit anything for 2d case at least that similar properties / paradox will be there?

Appendix B2
- proof 4.9 $f=gh$: $g(H)$ -> $g^\\#(H)$ in the last sentence
- proof 4.9 $f(x)=cx$: $X'$ should be internal interior of $H_inf(X)$ not "equal or internal interior".
- proof 4.9 relu: why ReLU$(m)$?
- proof 4.9 $x_1+x_2$: please polish sentence "Then $H_\inf(X)=$..." There is some typos as hull is not used in some places and strange notation on 2d case is used for the box with $r_a + r_b$ - this we cannot sum.

**Strengths And Weaknesses:**

Strength:
- Proved theorems and lemmas
- Figures which helped to follow the paper context
- Interesting results with trivial examples in the end to show problems with interval networks in context of robustness (this could be a reason of worse results for interval networks compared to classic ones)

Weakness:
- Changing notation and mixing different use cases throughout the paper make it hard to read and follow
- Some notations in Figures hard to follow
- Consideration of mainly 1d input space and problems with theorems for multi-dimensional inputs (here some clarification is needed from authors on comments below to be sure that everything holds in $\mathbb{R^n}$)
- Absence of any comments on flips dataset and paradox in multi-dimentional case (or how to inherit it from 1d case)

---

> ### Author Response · Authors · 2022-07-04
> **Response Part 1**
>
> We thank the reviewer for their many helpful comments and suggestions.  Below we have addressed these individually.  Many of these comments were typos, and if not addressed below they have simply been fixed in the revision.
>
> Importantly, we note immediately that our main results, Lemma 4.9 and Theorem 4.10 in fact do already apply to the multi-dimensional case, and explain more now.
>
> **Lemma 4.9 [Theorem 4.10] no any proof for multi-dimensional inputs and results cannot be simply extended to multi-dimensional case.**
>
> The theorem and proof is for the multi-dimensional case.  The “parallel” and “sum” cases handle higher-dimensional inputs, and the “duplication” case handles higher-dimensional outputs.  In the revision, we have made it clearer that the inductive definition of a neural network can handle taking multiple inputs and producing multiple outputs and have explained in detail how this translates into real vector spaces.  We furthermore create a distinction between the network program and network interpretation, allowing us to write a clearer example here:
> Take the two-layer deep residual neural network $n : \mathbb{R}^3 \rightarrow \mathbb{R}^2 $ $n(x_1,x_2,x_3) = (\text{ReLU}(x_1 + \text{ReLU}(x_2)) + \text{ReLU}(x_3), x_3)$.  In the new notation we’ve included with the paper, this might be encoded  as a network $n : (\mathbb{R} \times \mathbb{R}) \times \mathbb{R} \rightarrow \mathbb{R} \times \mathbb{R}$ written $\langle \mathsf{Sum} \circ \langle \text{ReLU} \circ \mathsf{Sum} \circ \langle 1\cdot, \text{ReLU} \rangle , \text{ReLU} \rangle, \mathsf{Sum} \circ \langle 0 , 1\cdot \rangle \rangle \circ \mathsf{Dup}$.  As these cases have each been covered by the proof, the theorem can handle multi-dimensional inputs.
>
> ## Strengths and Weaknesses
>
> **Changing notation and mixing different use cases throughout the paper make it hard to read and follow**
>
> The proof techniques for the general invertibility theorem and the 1-layer theorem are different enough that their presentation significantly benefits from a somewhat different notation. We have however attempted to unify the notation more in the revised version.
>
> **Some notations in Figures hard to follow**
>
> We have improved the notation in figures 1 and 3.
>
> **Consideration of mainly 1d input space and problems with theorems for multi-dimensional inputs (here some clarification is needed from authors on comments below to be sure that everything holds in $\mathbb{R}^n$)**
>
> Our main result, Theorem 4.10, is **not** restricted to 1-d input-spaces.  Pedagogically, we show a restriction of it to 1-d in corollary 5.2.
>
> **Absence of any comments on flips dataset and paradox in multi-dimentional case (or how to inherit it from 1d case)**
>
> It is true that our definition of the flip dataset is 1-d, and thus our theorems only immediately apply to this 1-d dataset, however a d-dimensional flip dataset where each dimension other than the first of every input is zero (we’ll call this a d-flip) also induces the same impossibility result.
>  This can be seen by reduction to Theorem 5.8:  Suppose there is a single layer ReLU network $f(x) = M * ReLU(Nx + b) + c$ that can provably $\alpha$-robustly classify k   d-flips.  Then we can construct a one-hidden-layer network $g(x’) = M * ReLU(NPx’ + b) + c$ that $\alpha$-robustly classifies k 1-flips. Here we let $P \in \mathbb{R}^{k\times 1}$ be $P_{1,1} = 1$ and $P_{j,1} = 0$ for $j > 1$.  By Theorem 5.8, k must be less than $\lceil\frac{2}{\alpha} \rceil + 5$.
> We have extended proposition 5.9 to the multi-dimensional case, and we believe 5.8 could be extended to further generalized datasets.
>
> ## General
>
> **I have never seen such notation for "forall" with dots between expressions.**
>
> We are happy to change it to a comma, but would note that this is not a particularly uncommon notation.  This notation is popular enough that it appears on the [wikipedia page](https://en.wikipedia.org/wiki/Universal_quantification#As_adjoint) for “universal quantification” numerous times in the section “as adjoint.” It was historically introduced in logic texts as a way to fix scoping and remove difficult to read parenthesis in the general case, although seems to now appear only in quantification.  It means roughly “place an open parenthesis here and a close parenthesis as far to the right as possible.”  It is sometimes known as the “church dot” as explained on [stackexchange](https://math.stackexchange.com/questions/311871/what-are-the-rules-for-the-use-of-dots-rather-than-parentheses-in-logical-formul) in detail (although for a more complicated and uncommon usage).
>
> ## Sec 3.1
>
> **I propose to remove notation $\mathbb{B}$ as it messes things with $\mathbb{R}$.**
>
> Using unstylized B can be confusing as we frequently use A,B,C as set examples.  We instead have changed the notation for the set of reals and naturals to use mathcal to be inline with $\mathscr{B}$ for the set of boxes in the revised paper.

---

> > ### Author Response · Authors · 2022-07-04
> > **Response Part 2**
> >
> >
> > ## Sec 3.2
> >
> > **first paragraph:  $\mathbb{B}_\epsilon(x)$ -> $\mathbb{B}^0_\epsilon(x)$. Even I think everywhere we can skip  in the superscription as it is not used in any other context.**
> >
> > Here we do in fact mean the ball including its surface.  We need this interior notation in section 4.1, and we use both in proofs in 4.2, and the surface included version in 5.1.
> >
> > **Maybe I missed, but I think Def 3.1 and 3.2 are given only for 1d input in the end, it cannot give us definition on multi-dimensional inputs. The way we can get multi-dim is duplication, which doesn't create different coordinates.**
> >
> > This is incorrect.  Duplication creates networks with higher dimensional output, not input.  “Parallel” and “addition” handles higher-dimensional input.   All our proofs here apply to multi-dimensional inputs.  These definitions include every possible ReLU network, and our proofs would not work if they did not.
> >
> > **Usage of $x_1$, and $x_2$ is not clear and the whole notation is messed between 1d and $n$-d cases. So it should be clarified, e.g. if  is a vector or not, from which space it is.**
> >
> > Whether $x$ is a vector or not depends on which case of the sigma-network definition the function falls into.  We have clarified this in the revision.
> >
> > **Def 3.2 Parallel: maybe add that $B=(x_1,x_2)$?**
> >
> > We have that $B=B|_1 × B|_2$.
> >
> > **Def 3.2 Constant: what is $m$, $n$?**
> >
> > You are correct, these no longer apply and we have updated the paper to reflect this.
> >
> > ## Sec 4.1
> >
> > **For the def 4.1 note why do we need to have center point in $relint(S)$?**
> >
> > Because if $S$ is a single point, its interior is empty but its relative interior is not.
> >
> > **last paragraph - could authors add any examples for both cases?**
> >
> > Yes, we will add examples in the next revision before the discussion period has ended.
> >
> > ## Sec 4.3
> >
> > **I found it is hard to read with changed notation on $x$**
> >
> > We have fixed this notation in the revision.
> >
> > **Th. 4.10: function takes $\mathbb{R}^n$ but in proof there is no justification for this.**
> >
> > The “parallel” and “sum” cases cover multi-dimensional input in this proof.  We realize that implicitly equivocating the more structured (but still multidimensional) spaces that our low-level operational definition of neural networks induced and d-dimensional real vector spaces may have introduced confusion to make this more explicit.  We have fixed this in the updated version to make this process explicit.
> >
> > **I like here example of fig. 3 to demonstrate the outcome from the theorem, but it is not clear how to use this theorem in general, and why it is about imprecision of interval? Maybe some extended comment / clarification, or even abstract explanation in case  is some set of the points which is mapped to the same  (so non-invertible on this point).**
> >
> > Corollary 5.2 is meant to provide an example which motivates the importance of this theorem.
> >
> > ## Sec. 5
> >
> > **Corollary 5.2 what about other $\nu$?**
> >
> > Theorem 4.10 and by extension corollary 5.2 only work for $\nu=1$.  When relaxing to $\nu < 1$, theorem 5.8 is applicable.
> >
> > **Def. 5.3 for $F$ is given not clearly.**
> >
> > We are unsure what is unclear about this definition.  Can you please specify?
> >
> > **The whole section seems weak to me as results only for 1d input which is not much interesting. Can we inherit anything for 2d case at least that similar properties / paradox will be there?**
> >
> > Theorem 5.8 can be trivially extended to flips with extra unused dimensions.  We added this in the revision as corollary 5.9. Furthermore, we have included an extension of (what is now) proposition 5.10 to the multi-dimensional case, as proposition 5.11.
> >
> > ## Appendix B2
> >
> > **There are some typos as hull is not used in some places and strange notation on 2d case is used for the box with $r_a + r_b$ - this we cannot sum.**
> >
> > In 1 dimension as we have here, we simplify the “hull” to an explicit definition of the ball that the hull produces.  Summing two balls is perfect when they are each 1 dimensional.

---

> > > ### Comment · Reviewer_f5Ug · 2022-07-09
> > > **Comments on the revision and authors' clarifications**
> > >
> > > I would like to thank authors for going through all my comments, all clarifications and new clearer version of the paper.
> > > All changes on notation and stronger formulation of spaces made paper much more readable and simpler to understand now.
> > >
> > > > Revision in Lemma 4.9 and Theorem 4.10 for multi-dimensional case
> > >
> > > Thanks for the formal definition and spaces specification. Now it is very clear for me and simple to read and understand where $x_1$ is the first coordinate of multi-dimensional vector or it is first component of operation and can be multi-dimensional vector.
> > >
> > > > Revision on "Absence of any comments on flips dataset and paradox in multi-dimentional case (or how to inherit it from 1d case)”
> > >
> > > Thanks, now it is very clear what you meant and I like simple example. In future I wonder to see any truly multi-dimensional dataset (that data’s manifold is not 1d) construction for which we can have same negative result. But for current work I agree the proofs and examples fulfilled the claimed results.
> > >
> > > > Comment on “I have never seen such notation for "forall" with dots between expressions.”
> > >
> > > Wow, thanks for references. I was working on PDE and functional analysis and for some reason we never used this notation. I am good to have church dot notation.
> > >
> > >
> > > Overall, I am happy with current version and justifications on the multi-dimensional case. I did another pass and have several suggestions to improve a bit further. Also I did more careful pass on Section 5 having now clearer notation (comments on improving are also below).
> > > - Sec 2, typo "significant gab" -> "significant gap"
> > > - Sec 3.1 first paragraph "For a set $Y$" - from which space elements in $Y$ are from? Maybe add $Y=\\{y. y\in T_1 \times T_2 ... \times T_n\\}$ and $Y|_i = \\{y_i\in T_i. y\in Y\\}$ and move this definition to the end of second paragraph. Then it is clear how do you use restriction in different contexts: either i-th dimension and this is scalar or the i-th space in case of product of spaces.
> > > - Sec 3.1 propose to add with notation $\mathcal{B}^0_\epsilon$ also $\mathcal{B}_\epsilon$ as it is not clear that it is "mean the ball including its surface"
> > > - Sec 3.2 "a proof of robustness at a point is does" -> "a proof of robustness at a point does" (maybe I again misunderstood English)
> > > - Sec 4.1 "using notion of relative interior" -> "using known notion of relative interior"
> > > - Sec 4.2 proof for $<g_1, g_2>$: $H_\inf(g_1(X|_2))$ -> $H_\inf(g_2(X|_2))$, no? (same in Appendix)
> > > - Sec 4.3 "assuming $M$ is a box" - just add $M\in\mathcal{B}^S$ (your variant of B :) )
> > > - Sec 5.2 first paragraph: it is not clear expression "do not touch" - what do not touch?
> > > - Def 5.3 what is $\mathcal{P}$?
> > > - Corollary 5.9: $M\in\mathbb{R}^{d\times 1}$ -> $M\in\mathbb{R}^{1\times d}$ or use transpose, otherwise strange notation on matrix vector multiplication.
> > > - Sec 5.3 "define an neural" -> "define a neural"
> > > - Appendix A lemma 5.6 - why use inequality while it should be equality for derivative, e.g. $f'(l)\leq -1$ -> $f'(l)= -1$
> > > - Appendix A, last row before Lemma 5.7 - closing $\\}$ bracket is missed
> > > - Appendix A lemma 5.7. Why $- a^{-1} * 2 *f(\hat{x}_1) \leq -c$ or $f(\hat{x}_1)  \geq 1$? This is used to get second inequality in proof. It is even used as equality in the main text for some reason. I didn't get this.
> > > - Appendix A lemma 5.7. Also there is mismatch between inequalities for $f'(l) - f'(u)$ in Appendix proof and main text comments. Could authors add more precise formulas for this estimation as I didn't get this estimation at all (after we use formulas for derivatives).
> > > - Appendix A theorem 5.8. Typo: $Mc_{\alpha,j}+2d=M(v_{\alpha,j}^+ + v_{\alpha,j}^+)$ -> $Mc_{\alpha,j}+2d=M(v_{\alpha,j}^+ + v_{\alpha,j}^-)$. I think better to use strict equality where it is possible (simpler to read the proof). Could authors clarify why we know (from where) that $(-1)^j Mc_{\alpha,j}+2d \geq |M| w_{\alpha,j}$ (so the value of function on the boarder of the interval we mapped is smaller than in the center? Sorry, I didn't get this. We only know that it has the same sign but we don't know about exact value, I believe).
> > >
> > > The rest of the proofs looks correct to me in case authors can clarify several inequalities in above comments.
> > >
> > > One more question for my understanding. Could it be that deeper NNs are imperfect provable robust? As I understood the answer is yes and it is done in Baader et al.

---

> > > > ### Comment · Reviewer_f5Ug · 2022-07-12
> > > > **Any questions?**
> > > >
> > > > Dear authors,
> > > >
> > > > Feel free to ask if you have any questions regarding my last comments and suggestions.

---

> > > > > ### Author Response · Authors · 2022-07-12
> > > > > **Thanks for the suggestions!**
> > > > >
> > > > > Thanks for the thorough suggestions and comments!  We are working to revise the paper to incorporate these.

---

> > > > ### Author Response · Authors · 2022-07-13
> > > > **Fixed suggestions**
> > > >
> > > > > Sec 2, typo "significant gab" -> "significant gap"
> > > >
> > > > Fixed.
> > > >
> > > > > Sec 3.1 first paragraph "For a set $Y$" - from which space elements in $Y$ are from?
> > > >
> > > > Fixed.
> > > >
> > > > > and move this definition to the end of second paragraph.
> > > >
> > > > We believe that this definition with the inclusion of spaces meant is general and easy to read enough that it can go into a "general notation" paragraph coming first unrelated to the quirks of how our paper discusses boxes.
> > > >
> > > > > Sec 3.1 propose to add with notation ...
> > > >
> > > > We have swapped the definition of the interior of the ball with the prior sentence to make this clearer.
> > > >
> > > > > Sec 3.2 "a proof of robustness at a point is does" -> "a proof of robustness at a point does" (maybe I again misunderstood English)
> > > >
> > > > Fixed.
> > > >
> > > > > Sec 4.1 "using notion of relative interior" -> "using known notion of relative interior"
> > > >
> > > > Updated.
> > > >
> > > > > Sec 4.2 proof
> > > >
> > > > Fixed.
> > > >
> > > > > Sec 4.3 "assuming $M$ is a box"
> > > >
> > > > Fixed.
> > > >
> > > > > Sec 5.2 first paragraph: it is not clear expression "do not touch" - what do not touch?
> > > >
> > > > The robust regions around inputs.  It should have said input points, not boxes.  Hopefully this is clearer.
> > > >
> > > > > Def 5.3 what is $\mathcal{P}$?
> > > >
> > > > Just as $\sharp$ has no meaning on its own, in this context $\mathcal{P}$ alone is meaningless.  $f^\mathcal{P}$ is the perfect transformer.  Occasionally the symbol for the set of points in the domain is used instead of, or in addition to $\sharp$ to denote the type of transformer, for example $f^\sharp_\mathscr{B}$ or $f^\mathscr{B}$ for the interval transformer.  In this setting, the domain of possible sets is the powerset of the domain of possible values of concrete functions. $\mathcal{P}$ is used in the rest of the paper for powerset, so we considered this an appropriate name and notation.    We have added this definition in background.
> > > >
> > > > > Corollary 5.9
> > > >
> > > > It is correct as written.  $f' : \mathbb{R} \rightarrow \mathbb{R}$ and $f : \mathbb{R}^d \rightarrow \mathbb{R}$.  This means $x \in \mathbb{R} \simeq \mathbb{R}^{1 \times 1}$ and typically matrix multiplication is written $MN \in \mathbb{R}^{a\times b}$ for $M \in  \mathbb{R}^{a \times m}$ and $N \in  \mathbb{R}^{m \times b}$.  Here we have $M \in \mathbb{R}^{d \times 1}$ times $x \in \mathbb{R}^{1 \times 1}$.
> > > >
> > > > > Sec 5.3 "define an neural" -> "define a neural"
> > > >
> > > > Fixed.
> > > >
> > > > > Appendix A lemma 5.6 - why use inequality while it should be equality.
> > > >
> > > > We are deducing a weakened claim from the MVT and not simply restating the theorem. This should make it clearer how to connect to the statements that follow.
> > > >
> > > > > Appendix A, last row before Lemma 5.7 - closing
> > > >
> > > > Fixed.
> > > >
> > > > > Appendix A lemma 5.7. Why $-a^{-1} * 2 * f(\hat x_1) \leq -c$
> > > >  or $f(\hat x_1) \geq 1$? This is used to get second inequality in proof. It is even used as equality in the main text for some reason. I didn't get this.
> > > >
> > > > This was a typo.  The inequality should have $-c$ in the appendix and $-\hat l_j c$ in the main body.  This has been fixed and propagated.
> > > >
> > > > > Could authors add more precise formulas for this estimation as I didn't get this estimation at all.
> > > >
> > > > We don't use the convenience notation defined formally in the appendix in the main-body proof overview.  The meaning is the same.  These are not really estimations, but can be named as they look like derivative estimations.
> > > >
> > > > > Appendix A theorem 5.8. Typo
> > > >
> > > > Fixed.
> > > >
> > > > > Could authors clarify why we know (from where) that ...
> > > >
> > > > We know $f^\sharp(\langle \hat x_j, \alpha \rangle)=\langle \frac{1}{2}Mc_{\alpha,j} + d, \frac{1}{2}|M|w_{\alpha,j} \rangle$, and that $\hat l_j \in f^\sharp(\langle \hat x_j, \alpha \rangle)$ and
> > > > by contradiction assumption that $0 \notin f^\sharp(\langle \hat x_j, \alpha \rangle)$.  Thus, $|c|-|r| > 0$ and thus $|c|-r > 0$ and thus $2|c|-2r > 0$ .  We know $c = \frac{1}{2}Mc_{\alpha,j} + d$ and $r = \frac{1}{2}|M|w_{\alpha,j}$  so $2|c|>2|r|$ and thus $|Mc_{\alpha,j} + 2d| > |M|w_{\alpha,j}$ and we know $sign\;c = sign\; \hat l_j$ so $|Mc_{\alpha,j} + 2d| = \hat l_j (Mc_{\alpha,j} + 2d)$ and thus  $(-1)^j(Mc_{\alpha,j} + 2d) \geq |M|w_{\alpha,j}$.
> > > >
> > > > > One more question for my understanding. Could it be that deeper NNs are imperfect provable robust?
> > > >
> > > > Yes, 2-hidden-layer neural networks can even be imperfectly provably robust.

---

> > > > > ### Comment · Reviewer_f5Ug · 2022-07-14
> > > > > **Seems updated revision is old.**
> > > > >
> > > > > Dear authors,
> > > > >
> > > > > Thanks for the comments and new revision. Could you recheck that you uploaded the correct version as I don't see any differences with between 13 July and 3 July versions. Maybe I missed something.
> > > > >
> > > > > Thanks!

---

> > > > > > ### Author Response · Authors · 2022-07-14
> > > > > > **Apologies for the old pdf in the update**
> > > > > >
> > > > > > We have re-tried to upload the revision, and it appears that it worked this time.

---

> > > > > > > ### Comment · Reviewer_f5Ug · 2022-07-17
> > > > > > > **Final comments (minor :) )**
> > > > > > >
> > > > > > > Dear authors,
> > > > > > >
> > > > > > > Yep, it works now.
> > > > > > >
> > > > > > > Thanks for all fixes and explanations. Now looks very clear to me and all my doubts are resolved. Several last minor comments:
> > > > > > >
> > > > > > > - Regarding explanation for theorem 5.8 - I would recommend to include explanation you gave to me in Appendix (maybe it is a bit obvious, but still clarify contradiction, at least include that zero is not in the interval from which we can get $|c| - |r| > 0$ and from this final equation).
> > > > > > > - Regarding Lemma 5.7 in the main text - I think it is still not precisely correctly written (but doesn't change any results). I believe it should be (having that $\hat{l}_j=f(\hat{x}_j)$): $\hat{l}_ja^{-1}(f(\hat{x}_j + a) - f(\hat{x}_j - a) = -\hat{l}_j \left[\frac{f(\hat{x}_j) - f(\hat{x}_j - a)}{a} - \frac{f(\hat{x}_j + a) - f(\hat{x}_j)}{a} - \hat{l}_jc\right] = -\hat{l}_j \left[\frac{f(\hat{x}_j) - f(\hat{x}_j - a)}{a} - \frac{f(\hat{x}_j + a) - f(\hat{x}_j)}{a}\right] +c$
> > > > > > > - Regarding Corollary 5.9: you write $d$-flip, that is why $x\in\mathbb{R}^d$ not $x\in\mathbb{R}^{1\times 1}$. And then $Mx$ is piece-wise multiplication.

---

> > > > > > > > ### Author Response · Authors · 2022-07-19
> > > > > > > > **Thanks for the comments!**
> > > > > > > >
> > > > > > > > > Regarding explanation for theorem 5.8 - I would recommend to include explanation you gave to me in Appendix (maybe it is a bit obvious, but still clarify contradiction, at least include that zero is not in the interval from which we can get  and from this final equation).
> > > > > > > >
> > > > > > > > We have done this in the newest revision.
> > > > > > > >
> > > > > > > > > Regarding Lemma 5.7 in the main text
> > > > > > > >
> > > > > > > > Yes, we have now updated this.
> > > > > > > >
> > > > > > > > > Regarding Corollary 5.9: you write $d$-flip
> > > > > > > >
> > > > > > > > $f$ takes $d$-flips yes, but $x$ isn't the input to $f$ it is the input to the function we are trying to construct $f'$, which only takes 1 dimensional points as input.  $M$ converts from the one dimensional input of $f'$ to d-dimensional input to $f$.  We will make this more clear.

---

> > > > > > > > > ### Comment · Reviewer_f5Ug · 2022-07-24
> > > > > > > > > **Checked the updates**
> > > > > > > > >
> > > > > > > > > Dear authors,
> > > > > > > > >
> > > > > > > > > Thanks for the updates :)!
> > > > > > > > >
> > > > > > > > > Sorry, had a mistake in previous comments and I see now that the last version in Lemma 5.7 needs only the fix of sign in the last term. Please, recheck and confirm that you agree with $\hat{l}_ja^{-1}(f(\hat{x}_j + a) + f(\hat{x}_j - a)) = -\hat{l}_j \left[\frac{f(\hat{x}_j) - f(\hat{x}_j - a)}{a} - \frac{f(\hat{x}_j + a) - f(\hat{x}_j)}{a} - c\right] = -\hat{l}_j \left[\frac{f(\hat{x}_j) - f(\hat{x}_j - a)}{a} - \frac{f(\hat{x}_j + a) - f(\hat{x}_j)}{a}\right] + \hat{l}_j  c$.
> > > > > > > > >
> > > > > > > > > Don't have any other comments, and I am happy with the final version. Thanks again for all the work and incorporating all my comments and advises.

---

> > > > > > > > > > ### Author Response · Authors · 2022-07-27
> > > > > > > > > > **Fixed**
> > > > > > > > > >
> > > > > > > > > > Yes, we believe you are correct and have updated the version.
> > > > > > > > > >
> > > > > > > > > > Thanks again for the comments!

---

### Author Response · Authors · 2022-07-03
**First revision after initial feedback.**

We thank reviewers for their thoughtful comments and suggestions.  We have posted an initial revision which addresses these comments and fixes the various typos and ambiguities discovered. In particular, multiple comments were made as to applicability of the results to higher-dimensional input.  As explained in detail in comments, our results for one-layer and standard robustness extend to the multi-dimensional cases, and we will update this in the revision.  Furthermore, we reiterate that our central result, Theorem 4.10, necessarily applies to both multi-dimensional input and output.  We have improved the presentation of the theorem to make it clearer how it attains its result for multi-dimensional inputs, in particular by clarifying the kinds of spaces that our inductive networks operate on.  We show in a comment to reviewer f5Ug how a multi-dimensional neural network can be represented using just the simple operations of our inductive network definition.

In the comments here, we attempt to use the updated notation from the paper.  Furthermore, we will include a symbol table as suggested by Reviewer N9Hb.

---

### Decision · Action_Editors · 2022-07-27

**Recommendation:** Accept with minor revision

**Comment:**

The paper received positive feedback from the reviewers and the authors have responded to earlier concerns of the reviewers. The authors are strongly advised to take into account detailed feedback from Reviewer f5Ug for preparing the final version.

All reviewers view the paper positively and therefore, the paper can be accepted with minor revision. Congratulations to authors on a solid work and I look forward to receiving the final version of the paper.


Best,

---

> ### Author Response · Authors · 2022-08-11
> **Thanks!**
>
> We thanks the reviewers and action editors for the helpful feedback!  We've uploaded a camera ready version that has taken into account all feedback.
>
> Best,
> Authors